# Understanding Multimodal Learning: A Loss Landscape Smoothness Perspective

## Abstract

A surge of recent advancements has consistently highlighted the superiority of multimodal learning over unimodal approaches across a variety of tasks. However, the theoretical foundations elucidating this advantage remain underexplored: existing theoretical analyses are often constrained by tight assumptions, and lack empirical validation. In this paper, we bridge this gap by proposing a novel theoretical framework grounded in *convolutional smoothing*, offering a new perspective on how multimodal learning contributes to a smoother loss landscape compared to unimodal learning. Building upon this theoretical foundation, we introduce a simple yet effective distributional training strategy based on stochastic modality pairing instead of a fixed pairing; thus, further promoting a flatter landscape via convolutional smoothing. Our empirical results across various multimodal datasets demonstrate that multimodal models not only achieve higher performance but also exhibit flatter loss landscape, which represent better generalization and robustness.

## 1 Introduction

Multimodal learning has emerged as a crucial deep learning approach, driven by the increasing demand to integrate diverse modalities of data sources. At its core, multimodal learning aims to combine complementary representations of multiple modalities, such as vision, language, and audio, to achieve richer and better representations. Early approaches primarily focus on fusing modality-specific features either through simple concatenation or via projecting them into a shared latent space. Building upon these foundations, advanced methods have shown meaningful improvements over unimodal baselines by independently regulating the learning rates for each modality (Fujimori et al., 2020; Yao & Mihalcea, 2022) or modulating gradients between modalities (Peng et al., 2022; Li et al., 2023). Concurrently, the recent establishment of large-scale multimodal datasets has been accelerating the progress of large-scale models (Vaswani et al., 2017; Dosovitskiy et al., 2021), most notably in the CLIP-based models (Radford et al., 2021). These advancements continue playing a pivotal role in pioneering the era of Foundation Model (Achiam et al., 2023; Girdhar et al., 2023).

Alongside these empirical advances, efforts to theoretically understand the main advantages of multimodal learning have been made. A prior theory argues that leveraging multiple modalities can improve training efficiency and generalization by providing a more comprehensive coverage of the latent feature space compared to unimodal learning (Huang et al., 2021). Alternatively, a recent work offers a mathematical foundation, proving that successful multimodal representation learning is possible when the input data exhibits sufficient heterogeneity and consistency across modalities (Lu, 2023). In addition, one of the very recent works has theoretically proposed that one modality model can benefit synergistically from other modality models by aligning with their feature distributions, even without access to the exactly paired annotations across modalities (Lee & Yoon, 2025).

However, these prior theories have clear limitations in providing a comprehensive understanding, with an apparent coincidence between theory and experiments. They are based on scarce empirical verification on synthetic data and simple datasets, or strong theoretical assumptions for the existence of an ideal representation across modalities. Moreover, prior approaches often handle the generalization bounds or Rademacher complexity analyses; they frequently rely on strong and sometimes restrictive assumptions (Huang et al., 2021; Lu, 2023), such as Lipschitz continuity, which may not hold in practical deep learning settings. Thus, some empirical studies even raise a doubt that multimodality sometimes hinders performance due to challenges such as information overload, leading to overfitting

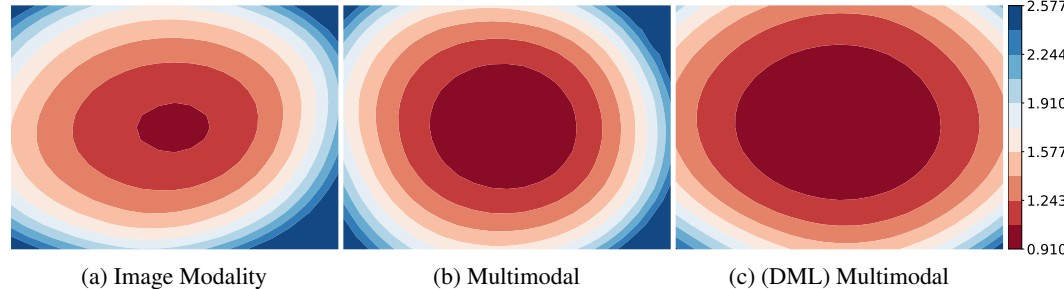

|  (a) Image Modality | (b) Multimodal | (c) (DML) Multimodal |

Figure 1: **Loss landscape visualizations for the AVMNIST dataset.** This figure presents a 2D projection of the loss landscape, following the loss landscape visualization proposed in (Li et al., 2018). Here, DML (Figure 1c) stands for Distributional Multimodal Learning (see Section 3.3).

or modality competition (Wang et al., 2020; Huang et al., 2022). To address this limitation, we advocate for a more general theoretical perspective that can naturally extend to modern deep learning.

Hence, we alter into a novel perspective beyond these conventional theoretical points of view, by introducing an alternative interpretation of multimodal learning through the lens of the flatness of the loss landscape. Specifically, we propose that the presence of additional modalities introduces a form of *convolutional smoothing* over the loss landscape associated with unimodal learning. In a nutshell, for a data $\mathbf{x}_i$ of modality $i$ paired with the paired $\mathbf{x}_j$ of modality $j$, multimodal learning naturally smooths the loss surface by minimizing the expected loss over the joint probability $p(\mathbf{x}_i, \mathbf{x}_j)$, which is shown to be analogous to a convolution process. This smoothing effect reduces the sharpness of the landscape, effectively guiding the optimization process toward flatter minima, which are often associated with better robustness. Notably, this behavior arises intrinsically from the characteristics of multimodal training itself, without explicitly applying techniques designed to promote flatness, highlighting as the novel perspective. Building on this insight, we additionally introduce simple technical approach called *Distributional Multimodal Learning* (**DML**), that leverages the smoothing effect by randomly pairing samples across modalities $i$ and $j$ within the same hypothesis space.

To offer an intuitive teaser, we provide a preliminary result that visualizes the loss landscape. As shown in Figure 1, multimodal learning on the AVMNIST dataset with image (vision) and audio modalities indeed produces significantly smoother and flatter loss landscapes compared to unimodal learning. Moreover, DML tends to converge to flatter minima than standard multimodal learning, in which the input data are strictly paired one-to-one; DML promotes this smoothing effect through stochastic pairing of the data. Taken together, these preliminary results highlight the potential of our theoretical framework to advance multimodal learning as a means of achieving more robust and generalizable outcomes than unimodal approaches, particularly under diverse modality pairings.

## 2 RELATED WORKS

**Late Fusion Approach in Multimodal Learning**   Multimodal learning aims to enhance model performance by integrating information from different modalities, such as vision, audio, and text. Among the various fusion strategies, late fusion and early fusion have gained significant attention in recent works (Ramirez et al., 2011; Baltrušaitis et al., 2018). On late fusion, each modality is processed independently through dedicated encoders and fused at the late stage, where it preserves modality-specific representations, supports flexible alignment in a shared latent space Early fusion focuses on combining input-level data from multiple modalities, allowing joint processing from the beginning of the model, which can potentially capturing deeper interactions between modalties. Recent methods enhance this strategy by incorporating cross-modal attention and transformer-based to improve interaction across modalities. In this study, we adopt a late-fusion strategy. This design choice enables clearer analysis of the latent representations from each modality and allows a more proper comparison between the intrinsic properties of multimodal and unimodal learning.

**Flat Minima and Generalization/Robustness Performance**   Finding flatter minima has emerged as a main key concept for understanding robustness and generalization in deep learning. While better generalization is not exactly equivalent to finding flatter minima, substantial evidence suggests that flatter loss landscape often correlate with improved generalization and robustness to distribution

shifts. For instance, prior methods such as Stochastic Weight Averaging (SWA) (Izmailov et al., 2018) and its dense variant (SWAD) (Cha et al., 2021) employ heuristic approach for locating flatter regions of the loss landscape by averaging model parameter weights. More recent approaches, such as Sharpness-Aware Minimization (SAM) and its extensions (Foret et al., 2021; Kwon et al., 2021; Zhuang et al., 2022), cast training as a min-max optimization, explicitly minimizing the worst-case loss within a neighborhood of the current parameters and thereby converge to smooth region.

However, despite the importance of flatness, the role of loss landscape geometry in multimodal learning has received relatively limited attention, especially from a theoretical perspective. Some prior works have occasionally reported flatness metrics to highlight empirical gains (Wei & Hu, 2024), but lack in deep analysis of the underlying mechanisms why it leads to smooth loss landscape. Therefore, we aim to deepen this understanding by analyzing the behavior of multimodal learning from the perspective of loss landscape. Through both qualitative and quantitative results, we show that multimodal learning leads to flatter minima, which in turn contributes to improved robustness.

## 3 THEORETICAL VIEWS: CONVOLUTIONAL SMOOTHING OF MULTIMODAL

In this section, we provide the detailed theories of the concept of *convolutional smoothing*, which naturally appears in multimodal learning, leading to flatter minima than in unimodal learning.

### 3.1 SETUP AND ASSUMPTIONS

Before proposing main theories, we first introduce basic notations to be used in our analyses.

- $\mathbf{x}_i \in \mathcal{X}_i \subset \mathbb{R}^{d_i}, \mathbf{x}_j \in \mathcal{X}_j \subset \mathbb{R}^{d_j}$ denote input data from two different modalities $i$ and $j$, with their data distribution $\mathbf{x}_i \sim p(\mathbf{x}_i)$ and $\mathbf{x}_j \sim p(\mathbf{x}_j)$, respectively.

- $y \in \mathcal{Y}$ be the target label, which indicates the shared semantics across modalities.

- $f_i(\mathbf{x}_i), f_j(\mathbf{x}_j) \in \mathbb{R}^d$ are the modality encoders for the respective modality.

- $f(\mathbf{x}_i, \mathbf{x}_j; \theta) := \phi(f_i(\mathbf{x}_i), f_j(\mathbf{x}_j))$ be a function parametrized by $\theta \in \mathbb{R}^m$, with modality specific function $f_i(\cdot)$ and $f_j(\cdot)$, and modality fusion function $\phi(\cdot, \cdot)$

- $\ell(f(\mathbf{x}_i, \mathbf{x}_j; \theta), y)$ is the loss function.

To formally compare unimodal and multimodal learning, we introduce a fusion function $\phi$ whose goal is to combine the representations from different modalities. This function can take various forms, such as simple additive fusion or concatenation plus multi-layer perception (MLP), both of which are widely used in state-of-the-art multimodal learning approaches (Liang et al., 2021; Tsai et al., 2019).

We make the following assumption for approximating the translation-invariance property of the $\phi$:

**Definition 1** (Approximate Translation-Invariance (ATI)). *Let $\phi : \mathbb{R}^{d_i} \times \mathbb{R}^{d_j} \to \mathbb{R}^d$ be a mapping function. Then, we define that $\phi$ exhibits Approximate Translation-Invariance (ATI) if there exists a vector $\boldsymbol{\alpha} \in \mathbb{R}^d$ such that*

$$\phi(\mathbf{u}, \mathbf{v} + \boldsymbol{\tau}) = \phi(\mathbf{u}, \mathbf{v}) + \boldsymbol{\alpha} \odot \boldsymbol{\tau}$$

*for all $\mathbf{u}, \mathbf{v}, \boldsymbol{\tau} \in \mathbb{R}^d$, where $\odot$ denotes element-wise (Hadamard) product.*

**Assumption 1.** *Let $\phi$ be a modality fusion function. Then, we assume that $\phi$ satisfies the Approximate Translation-Invariance (ATI) property.*

**Assumption 2** (Conditional Independence). *With given $y$, the two modalities are conditionally independent: $p(\mathbf{x}_i, \mathbf{x}_j \mid y) = p(\mathbf{x}_i \mid y)p(\mathbf{x}_j \mid y)$.*

For example, additive fusion with setting $\boldsymbol{\alpha} = \mathbf{1}$ fully satisfies the assumption, since each modality contributes equally without distortion. Another example, in the case of concatenation with non-linear MLP, the term $\boldsymbol{\alpha} \odot \boldsymbol{\tau}$ can be interpreted as a first-order Taylor approximation derived from the Jacobian of $\phi$, reflecting how small perturbations propagate through the nonlinearity. Detailed case studies, including fusion networks mentioned on Assumption 1, have provided in Appendix D.1.

Next, we define the expected unimodal and multimodal losses as follows:

$$\mathcal{L}_{\text{uni}} = \mathop{\mathbb{E}}_{y \in \mathcal{Y}} \left[ \int \ell(f(\mathbf{x}_i, \bar{\mathbf{x}}_j; \theta), y) p(\mathbf{x}_i \mid y) \, d\mathbf{x}_i \right], \tag{1}$$

$$\mathcal{L}_{\text{multi}} = \mathop{\mathbb{E}}_{y \in \mathcal{Y}} \left[ \iint \ell(f(\mathbf{x}_i, \mathbf{x}_j; \theta), y) p(\mathbf{x}_i, \mathbf{x}_j \mid y) \, d\mathbf{x}_i \, d\mathbf{x}_j \right]. \tag{2}$$

where $\bar{\mathbf{x}}_j$ represents the absence of modality $j$ (e.g., zero vector, or constant). For simplicity, we denote each loss, $\mathcal{L}_{\text{uni}} = \mathop{\mathbb{E}}_{y \in \mathcal{Y}} [\mathcal{L}_{\text{uni}}(f \mid y)]$ for the expected unimodal loss and $\mathcal{L}_{\text{multi}} = \mathop{\mathbb{E}}_{y \in \mathcal{Y}} [\mathcal{L}_{\text{multi}}(f \mid y)]$ for the expected multimodal loss.

### 3.2 MAIN THEORETICAL APPROACH: CONVOLUTIONAL SMOOTHING EFFECT

Under assumptions of approximate translation invariance of $\phi$ and the condition of paired modalities, we introduce that the multimodal loss can be interpreted as a convolutional smoothing of the unimodal loss in this section. Before deriving the convolutional form of the multimodal loss, we first define the concepts of the fusion-induced scaled shift and its corresponding kernel.

**Definition 2** (**Fusion-Induced Scaled Shift and Kernel**). *Let $\bar{\mathbf{x}}_j$ denote input data representing absence of modality $j$, such zero vector or constant value, and define fusion-induced scaled shift as :*

$$\tau_{\boldsymbol{\alpha}}(\mathbf{x}_j) := \boldsymbol{\alpha} \odot \left( \phi(f_i(\mathbf{x}_i), f_j(\mathbf{x}_j)) - \phi(f_i(\mathbf{x}_i), f_j(\bar{\mathbf{x}}_j)) \right), \tag{3}$$

*The corresponding* scaled-shift kernel *is then defined using the Dirac delta function* $\delta_{\tau_{\boldsymbol{\alpha}}(\mathbf{x}_j)}(\cdot)$ *centered at* $\tau_{\boldsymbol{\alpha}}(\mathbf{x}_j)$:

$$\mathcal{K}_{\mathbf{x}_j, \boldsymbol{\alpha}}(\boldsymbol{\tau}) := \mathop{\mathbb{E}}_{\mathbf{x}_j \sim p(\mathbf{x}_j \mid y)} \left[ \delta_{\tau_{\boldsymbol{\alpha}}(\mathbf{x}_j)}(\boldsymbol{\tau}) \right], \tag{4}$$

We then present the following theorems, which rigorously establish that the multimodal learning loss can be expressed as a convolution of the unimodal loss with the scaled-shift kernel, $\mathcal{K}_{\mathbf{x}_j, \boldsymbol{\alpha}}$.

**Theorem 1** (**Convolutional Smoothing with Modality Scaled Shifted Kernel**). *Under the Assumption 1, the expected multimodal loss is a scaled convolution of the unimodal loss:*

$$\mathcal{L}_{multi} = \mathop{\mathbb{E}}_{y \in \mathcal{Y}} \left[ \left( \mathcal{L}_{uni} \circledast \mathcal{K}_{\mathbf{x}_j, \boldsymbol{\alpha}} \right) (f \mid y) \right]. \tag{5}$$

**Theorem 2** (**Smooth Loss Landscape of Multimodal Learning**). *Suppose that conditional unimodal loss $\mathcal{L}_{uni}(f \mid y)$ is continuous and bounded for all $y \in \mathcal{Y}$, and the shift kernel $\mathcal{K}_{\mathbf{x}_j, \boldsymbol{\alpha}}$ has finite variance. Then the $\mathcal{L}_{multi}$ is smoother than the $\mathcal{L}_{uni}$, in the following two folds:*

1. ***Upper Bound on the Spectral Norm of the Multimodal Hessian:*** *For all $f$, the largest spectral norm of the multimodal Hessian is upper-bounded by that of the unimodal loss:*

$$\sup_f \|\nabla^2 \mathcal{L}_{multi}(f \mid y)\|_2 \leq \sup_f \|\nabla^2 \mathcal{L}_{uni}(f \mid y)\|_2. \tag{6}$$

2. ***Frequency Domain Interpretation:*** *Let $\widehat{\mathcal{L}}_{multi}(\omega)$, $\widehat{\mathcal{L}}_{uni}(\omega)$, and $\widehat{\mathcal{K}}_{\mathbf{x}_j, \boldsymbol{\alpha}}(\omega)$ denote the Fourier transforms of $\mathcal{L}_{multi}$, $\mathcal{L}_{uni}$, and $\mathcal{K}_{\mathbf{x}_j, \boldsymbol{\alpha}}$ respectively. Then, Eq. 5 can be expressed in the frequency domain as: $\widehat{\mathcal{L}}_{multi}(\omega) = \widehat{\mathcal{L}}_{uni}(\omega) \cdot \widehat{\mathcal{K}}_{\mathbf{x}_j, \boldsymbol{\alpha}}(\omega)$. Moreover, it follows that:*

$$\mathcal{O}(\widehat{\mathcal{L}}_{multi}(\omega)) \leq \mathcal{O}(\widehat{\mathcal{L}}_{uni}(\omega)). \tag{7}$$

The proofs of Theorems 1 and 2 are provided in Appendix A. We now present Remarks 2.1 and 2.2, offering theoretical insights into the fundamental properties of multimodal learning:

**Remark 2.1** (**Worst-Case Smoothness Comparison (Eq. 6)**). *The **hessian spectral norm** is widely used to quantify the smoothness of the loss landscape (Ghorbani et al., 2019), where lower eigenvalues or spectral ratios indicates flatter region. In this context, our analysis establishes a key theoretical guarantee: the worst-case Hessian spectral norm of a multimodal loss, $\mathcal{L}_{multi}$ is upper-bounded by that of a unimodal loss, $\mathcal{L}_{uni}$. This view shows that **multimodal learning approach at least have smoother or equal loss landscape under worst-case conditions** compare to that of unimodal.*

**Remark 2.2** (**Multimodal as a Low-Pass Filter in the Frequency Domain (Eq. 7)**). *The smoothing mechanism of multimodal learning can be elucidated from a **frequency-domain** perspective: $\mathcal{L}_{multi}$ attains smoother minima due to the influence of the scaled-shift kernel $\widehat{\mathcal{K}}_{\mathbf{x}_j,\boldsymbol{\alpha}}$. In particular, when the underlying distribution of $\mathcal{K}_{\mathbf{x}_j,\boldsymbol{\alpha}}$ exhibits sufficient variance, its Fourier transform $\widehat{\mathcal{K}}_{\mathbf{x}_j,\boldsymbol{\alpha}}$ decays rapidly as the frequency magnitude $\|\omega\| \to \infty$. This decay acts as a low-pass filter that **effectively suppresses the high-frequency components** of $\widehat{\mathcal{L}}_{uni}(\omega)$. Consequently, $\mathcal{L}_{multi}$ generally exhibits a smoother and flatter loss landscape compared to $\mathcal{L}_{uni}$ in frequency domain perspective.*

To support the validity of our theoretical framework, we emphasize a key consideration: the latent representations $f_i(\mathbf{x}_i)$ and $f_j(\mathbf{x}_j)$ should exhibit meaningful differences. This aligns with prior theoretical work emphasizing the importance of modality heterogeneity, that modalities should be different, but not arbitrarily so—for effective multimodal learning (Lu, 2023). For instance, in extreme cases where one modality consists of random Gaussian noise, its contribution becomes negligible during training, making it practically irrelevant. To better illustrate these theoratical frameworks, we present a conceptual illustration of our framework in Figure 2.

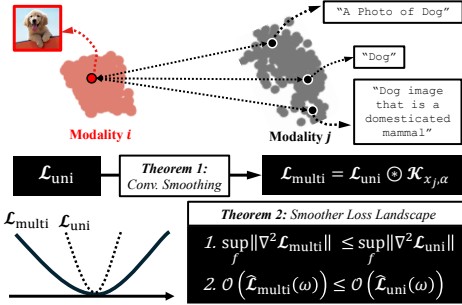

Figure 2: A conceptual sketch of our theoretical frameworks

### 3.3 BEYOND POINTWISE PAIRING: TOWARD DISTRIBUTIONAL MULTIMODAL LEARNING

Standard multimodal learning typically relies on pointwise-paired datasets, emphasizing exact correspondences between modalities under the empirical risk minimization framework. From a representation learning perspective, this approach leverages modality-specific information only at the level of individual data points. While it may implicitly capture certain aspects of modality-specific distributions, training remains limited as supervision is constrained to pointwise alignments.

To address this limitation, we propose a simple modification on pairing modality: shuffling datapoints across modalities within the same label space. This encourages stronger distributional alignment and is *directly motivated by our theoretical framework* based on given Assumption 2. We refer to this approach as **D**istributional **M**ultimodal **L**earning (**DML**), as illustrated in Figure 3. For clarity, we also refer to the conventional pointwise-paired setting as **S**tandard **M**ultimodal **L**earning (**SML**).

For example, let us consider a classification task with input sets $\{\mathbf{x}_{i,n}^0\}_{n=1}^N$ and $\{\mathbf{x}_{j,n}^0\}_{n=1}^N$ respectively from the modalities $i$ and $j$, where these sets are corresponding to the same class label (e.g., here, the superscipt indicates class 0). Instead of enforcing strict pairings like $(\mathbf{x}_{i,1}^0, \mathbf{x}_{j,1}^0)$, we allow randomized cross-modal pairing such as $(\mathbf{x}_{i,1}^0, \mathbf{x}_{j,k}^0)$, where $k \in [1, N]$ is sampled from datapoints in modality $j$ that share the same label. This distributional setting encourages the model to minimize the expected loss over a variety of pointwise pairings, thereby reducing the variance of the learned representations.

| Settings | Modality $i$ | Modality $j$ |
|---|---|---|
| SML (Pointwise) | | |
| DML (Distributional) | | |

Figure 3: Exactly Paired (SML) vs. Randomly Paired (DML) Setting

DML supports the model to align class-level distributions rather than relying on exact data-level correspondences. Specifically, this approach apparently more coincides with our theoretical insights than SML. Since DML stochastically pair during the training, DML implicitly increases the variance, resulting in a broader supervision. This aligns to our theoretical insights provided by Theorem 2, where increasing variance of data distribution leads to smoother loss landscape. In other words, DML could possibly maximize the variance of the kernel $\mathcal{K}_{\mathbf{x}_j,\boldsymbol{\alpha}}$ by sampling $x_j$ from the entire marginal distribution $p(x_j \mid y)$, which conceptually leads to highest entropy. Then this high-variance kernel is precisely what Remark 2.2 requires to function as a powerful low-pass filter, effectively smoothing the loss landscape by suppressing high-frequency components in perspective of Frequency domain.

Table 1: **Classification accuracies on multimodal datasets.** For each dataset, we denote $i$ modality as dominant if it outperforms the other modality. For datasets with three modalities, we denote $i$ modality as the first and second most dominant, respectively, based on their relative performance. All results are averaged over 3 times trials, with standard deviations reported.

| Datasets | Kinetics-Sounds | | AVMNIST | | CREMA-D | | UPMC-Food101 | |
|---|---|---|---|---|---|---|---|---|
| | Type | Accuracy | Type | Accuracy | Type | Accuracy | Type | Accuracy |
| Modality $i$ | [A] | $51.95 \pm 0.86$ | [I] | $64.98 \pm 0.16$ | [A] | $57.30 \pm 1.96$ | [T] | $86.38 \pm 0.11$ |
| Modality $j$ | [V] | $45.70 \pm 0.77$ | [A] | $41.84 \pm 0.24$ | [V] | $49.16 \pm 0.66$ | [I] | $64.68 \pm 0.28$ |
| SML | [V+A] | $62.95 \pm 0.51$ | [I+A] | $70.30 \pm 0.12$ | [A+V] | $59.62 \pm 1.09$ | [T+I] | $91.48 \pm 0.08$ |
| DML | [V+A] | $\mathbf{65.89 \pm 0.50}$ | [I+A] | $\mathbf{71.69 \pm 0.08}$ | [A+V] | $\mathbf{59.77 \pm 1.01}$ | [T+I] | $\mathbf{92.84 \pm 0.05}$ |

## 4 EXPERIMENTAL RESULTS

Here, we present experimental results comparing unimodal and multimodal learning through the lens of loss landscape, along with descriptions of the datasets, settings, and analyses of the results.

### 4.1 EXPERIMENTAL SETTINGS

**Multimodal Datasets** We conducted experiments and analyses on four different multimodal datasets: **Kinetics-Sounds** (Arandjelović & Zisserman, 2018), **AVMNIST** (Vielzeuf et al., 2018), **CREMA-D** (Cao et al., 2014), and **UPMC-Food101** (Wang et al., 2015). **Kinetics-Sounds** is an Audio-Visual [A+V] dataset derived from the Kinetics-400 dataset that contains 400 classes of Youtube video, selecting 31 human-action classes with 10-second video clips. **AVMNIST** is an Image-Audio [I+A] dataset comprising $1 \times 28 \times 28$ size of MNIST images, paired with 112×112 spectrograms generated from spoken digit audio (Jackson et al., 2018). **CREMA-D** is an Audio-Visual [A+V] dataset designed for speech emotion recognition within 6 classes, containing both facial and vocal expressions. In last, **UPMC-Food101** is an Image-Text [I+T] dataset featuring 101 food categories with corresponding recipe texts. The detailed descriptions are provided in Appendix B.1.

**Model Architectures** We follow standard model architectures from prior works (Peng et al., 2022; Wei & Hu, 2024) of multimodal learning, just slightly modifying them to each dataset and modality. For Kinetic-Sound, AVMNIST and CREMA-D, we use a ResNet-based architecture (He et al., 2016), where audio data is transformed into spectrogram images. For UPMC-Food101, we follow (Xu et al., 2025) and use a pretrained BERT model as the text encoder and a pretrained with ImageNet-1k (Krizhevsky et al., 2012) ResNet-18 model as the vision encoder.

**Hyperparameter Settings** We mostly followed the standard hyperparameter settings used in multimodal learning (Xu et al., 2025) and additionally tune for more careful training. Training is conducted for 70 epochs across all experiments. We apply the SGD optimizer (Eon Bottou, 1998) as done in the previous works, and partially replace to the Adam optimizer (Kingma & Ba, 2015). Additional hyperparameter, such as batch size, and detail descriptions are provided on Appendix B.2.

### 4.2 RESULTS: PERFORMANCE COMPARISON BETWEEN UNIMODAL AND MULTIMODAL

In this section, we present results from reproduced experiments comparing unimodal learning, SML, and our empirical suggestion DML approach across several benchmark datasets. For each dataset, we report classification accuracy for individual modalities, SML, and DML. The dominant modalities are chosen based on their unimodal performance, with $i$ representing the most dominant modality. Additionally, improvements are highlighted by color: **Green** for **SML** and **Blue** for **DML**.

Our findings based on Table 1 demonstrate the power of cross-modal learning, especially the proposed DML, which stochastically matches samples. For example, the combination of [A] and [V] in **Kinetics-Sounds** produces a substantial performance increase over single modalities. While [A] achieves 51.95% accuracy individually, the multimodal approaches yield substantial improvements: SML reaches 62.95% (**+11.00%**), and DML further advances to 65.89% (**+13.94%**). The gain of DML (**+2.94%** over SML) indicates effective exploitation of cross-modal distributional alignment.

A similar trend appears in **AVMNIST**, where the dominant modality [I] attains 64.98% accuracy. Fusing [I] and [A] with SML improves performance to 70.30% (**+5.32%**), and applying DML yields

an additional gain to 71.69% (**+6.71%**). This suggests that distributionally matched supervision benefits rather than exactly matched pair over given modalities. For **CREMA-D**, the benefits are more modest but still consistent. [A] provides the highest unimodal accuracy at 57.30%, SML elevates this to 59.62% (**+2.32%**). DML achieves 59.77% (**+2.47%**, and **+0.15%** over SML), a marginal yet slight improvement, and continues to surpass both individual modalities. This pattern indicates that multimodal learning consistently outperforms single modalities, and even under moderate modality complementarity, class-level stochastic alignment can extract incremental gains. Finally, in **UPMC-Food101**, [T] alone achieves a strong baseline of 86.38%, but its combination with [I] under SML boosts accuracy to 91.48% (**+5.10%**), and DML further increases performance to 92.59% (**+6.46%**).

These findings demonstrate that SML consistently outperforms the dominant unimodal baselines across all datasets. Furthermore, DML achieves additional gains by aligning modality distributions at the class level, thereby leveraging diverse information for each modality model. These results strongly support our theoretical view that multimodal learning improves performance through facilitating flatter and smoother optimization landscapes relative to unimodal learning.

### 4.3 COMPARISON WITH ADDITIONAL APPROACHES

Here, we conducted additional experiments to demonstrate how the DML approach can complement additional techniques, such as Data Augmentation (DA), and to highlight its superior novelty compared to recent flatness-based approaches.

**Data Augmentation (DA)**   Data augmentation (DA) is a widely used technique in deep learning and is known to have a smoothing effect (Yoo & Yoon, 2025). In this sense, DML may appear similar to DA, since the number of possible pairings increases during training and also have smoothing effect based on Theorem 2. However, DML clearly different from DA, and even complement. While DML forms pairs based on the hypothesis space, focusing on pairing mechanisms, DA focuses on augmenting the data itself. Moreover, since these approaches have different mechanism, they can be applied together. To validate this, we conducted experiments demonstrating their complementary effect. As Table 2 shows, DML outperforms SML with DA and achieves an additional gain of approximately **+1.3%** on Kinetic Sound and AVMNIST. These results indicate that DML not only shares certain benefits with DA but also complements it, providing distinct advantages. We further evaluate DML with diverse DA techniques, where results are provided in Appendix C.2.

Table 2: Performance with DA

| DA | KS | AVMNIST |
|---|---|---|
| SML | 62.95 | 70.35 |
| + J + C + F | 64.51 | 71.89 |
| DML | 65.89 | 71.69 |
| + J + C + F | **67.19** | **73.01** |

J: Color Jitter, C: Cropping, F: Flipping

**Comparison with Flatness Methods**   We compare our approach to recent flatness-based methods to evaluate how DML maintains novelty in multimodal learning relative to standard smoothing techniques. Specifically, we applied Sharpness-Aware Minimization (SAM) (Foret et al., 2021), one of the most widely used flatness approaches, to SML and compared its performance with DML. As shown in Table 3, DML consistently outperforms SML + SAM across most datasets, with the exception of CREMA-D. This demonstrates that DML not only benefits from a smoothing effect but also achieves superior performance compared to recent flatness approaches. From a performance perspective, DML remains competitive with current flatness-based approaches.

Table 3: Comparison with SAM

| Dataset | SML+SAM | DML |
|---|---|---|
| KS | 64.19 | **65.89** |
| AVMNIST | 70.91 | **71.69** |
| CREMA-D | **60.51** | 59.77 |
| U-Food101 | 91.76 | **92.84** |

Due to space constraints, we present additional experimental results and detailed discussions on DA and flatness approaches in combination with DML in Appendix C.2 and Appendix C.3, respectively

## 5   ANALYSIS

In this section, we present comprehensive evaluations by comparing the generalization and robustness performance of single modality and multimodality. Specifically, we analyze with consideration of two complementary perspectives: qualitative insights through visualizations of the loss landscape under parameter perturbations, and quantitative metrics commonly used to assess robustness performances.

Table 4: **Flatness and robustness across various multimodal datasets.** For all reported metrics, lower values ($\downarrow$) indicate better performance.

| KS | $\lambda_{\max}$ ($\downarrow$) | $\lambda^*$ ($\downarrow$) | $\mathcal{L}_{\mathrm{LPF}}$ ($\downarrow$) | $\Delta_{\mathrm{LPF}}$ ($\downarrow$) |
|---|---|---|---|---|
| Modality $i$: [A] | 13.8997 | 1.4284 | 2.1848 | 0.4781 |
| Modality $j$: [V] | 10.4048 | 1.5235 | 2.8545 | 0.2083 |
| SML | 9.9691 | 1.4287 | 1.6703 | 0.3097 |
| DML | **9.5732** | **1.1686** | **1.4637** | **0.1659** |

| AVMNIST | $\lambda_{\max}$ ($\downarrow$) | $\lambda^*$ ($\downarrow$) | $\mathcal{L}_{\mathrm{LPF}}$ ($\downarrow$) | $\Delta_{\mathrm{LPF}}$ ($\downarrow$) |
|---|---|---|---|---|
| Modality $i$: [I] | 7.9102 | 2.9235 | 0.9829 | 0.0138 |
| Modality $j$: [A] | 9.3102 | 1.5225 | 1.9924 | 0.0021 |
| SML | 6.2560 | 1.4338 | 0.9129 | 0.0121 |
| DML | **5.6220** | **1.2492** | **0.8083** | **0.0083** |

| CREMA-D | $\lambda_{\max}$ ($\downarrow$) | $\lambda^*$ ($\downarrow$) | $\mathcal{L}_{\mathrm{LPF}}$ ($\downarrow$) | $\Delta_{\mathrm{LPF}}$ ($\downarrow$) |
|---|---|---|---|---|
| Modality $i$: [A] | 10.4196 | 1.3016 | 2.6775 | 0.5937 |
| Modality $j$: [V] | **8.2353** | 1.9430 | 3.2975 | **0.0714** |
| SML | 9.2389 | **1.1461** | 2.5305 | 0.5447 |
| DML | 9.1712 | 1.6461 | **2.5212** | 0.5112 |

| Food101 | $\lambda_{\max}$ ($\downarrow$) | $\lambda^*$ ($\downarrow$) | $\mathcal{L}_{\mathrm{LPF}}$ ($\downarrow$) | $\Delta_{\mathrm{LPF}}$ ($\downarrow$) |
|---|---|---|---|---|
| Modality $i$: [T] | 8.6738 | 2.0203 | 1.8767 | 1.1044 |
| Modality $j$: [I] | 10.0044 | 1.4935 | 2.9904 | 1.2510 |
| SML | 8.3238 | 1.8015 | 0.4666 | 0.0937 |
| DML | **8.3068** | 1.5825 | **0.4353** | **0.0682** |

$\lambda^*$: The ratio of the max eigenvalue $\lambda_{\max}$ (1st) to the 5th largest eigenvalue $\lambda_5$. ($\lambda^* = \lambda_{\max}/\lambda_5$)
$\Delta_{\mathrm{LPF}}$: Discrepancy between original loss and LPF loss. ($\Delta_{\mathrm{LPF}} = \mathcal{L}_{\mathrm{LPF}} - \mathcal{L}$)

## 5.1 FLATNESS/SMOOTHNESS EVALUATION

Here, we employ two widely recognized metrics of flatness to evaluate generalization and robustness: the *Hessian maximum eigenvalue* which reflects the local curvature of the loss surface, and the *Low-Pass Filter (LPF) metric* that quantifies the stability of the trained model under perturbations.

**Hessian Spectrum** The *Hessian maximum eigenvalue* (Ghorbani et al., 2019; Yao et al., 2020) quantifies the sharpest curvature direction of the loss landscape, with smaller values indicating flatter minima. Yet, unimodal and multimodal models differ in architecture and parameter count, direct comparison of absolute Hessian eigenvalues could be misleading. To facilitate a more comparable assessment, we additionally report the ratio between the largest and 5$^{\text{th}}$ ($\lambda_{\max}/\lambda_5$), a commonly used surrogate for the smoothness of loss curvature (Jastrzebski et al., 2020; Foret et al., 2021). Specifically, this ratio captures the anisotropy of the curvature and quantifies the local shape of the loss landscape; lower values indicate a flatter and more isotropic landscape.

**LPF Metric** The *LPF metric* (Bisla et al., 2022) complements this analysis by quantifying the sensitivity of the loss surface to perturbations via Gaussian smoothing. Specifically, we define $\Delta_{\mathrm{LPF}}$ as the difference between the original loss $\mathcal{L}$ and the smoothed loss $\mathcal{L}_{\mathrm{LPF}}$, obtained by convolving the loss landscape with a Gaussian filter. This discrepancy reflects the susceptibility of the loss landscape to perturbations: a smaller $\Delta_{\mathrm{LPF}}$ indicates greater robustness, as model's performance is less affected.

**Flatness Analysis** We compare these generalization and robustness metrics across three settings: the dominant unimodal modality, SML, and DML. As expected, the results on Table 4 largely align with our theoretical hypothesis—multimodal approaches generally lead to smoother loss landscapes and improved robustness. Prominently, DML consistently outperforms other methods across most multimodal datasets. For example, results on Kinetics-Sounds, AVMNIST and UPMC-Food101 datasets outperformed compare to all baselines, where multimodal approaches obviously shows lower flatness values compare to unimodal. Moreover, DML also usually shows lower values on given metrics compare to SML, where training with stochastically matched pairs reach to smoother surface.

A slight divergence from the overall trend appears on the CREMA-D dataset. The modality $j$, which is visual modality, shows marginally smaller values of $\lambda_{\max}$ and $\Delta_{\mathrm{LPF}}$, which would typically indicate a flatter and more robust loss surface. However, this modality still exhibits a relatively larger $\mathcal{L}_{\mathrm{LPF}}$, suggesting weaker predictive performance. Its $\lambda^*$ is highest among all modalities, implying larger gap between the dominant and subordinate Hessian eigenvalues and thus more curvature. In contrast, as with the other datasets, both $\lambda_{\mathrm{LPF}}$ and $\lambda^*$ decrease under multimodal learning, indicating that multimodal learning still converges to *flatter and optimal minima* relative to the unimodal learning.

**Loss Perturbation Sensitivity** We evaluate the flatness of the local loss landscape that has been proposed on SWAD (Cha et al., 2021) : $\mathcal{F}_{\mathrm{gap}}(\theta) = \mathbb{E}\left[|\mathcal{E}(\theta') - \mathcal{E}(\theta)|\right]$, which quantify changes in empirical risk under random perturbations to the model parameters, and also serves as qualitative results of landscape smoothness in visualization. Here, $\mathcal{E}(\theta)$ denotes the empirical risk at the original parameters $\theta$, and $\mathcal{E}(\theta')$ denotes the risk at perturbed parameters $\theta'$, where $\|\theta'\| = \|\theta\| + \epsilon$ and $\epsilon$ is

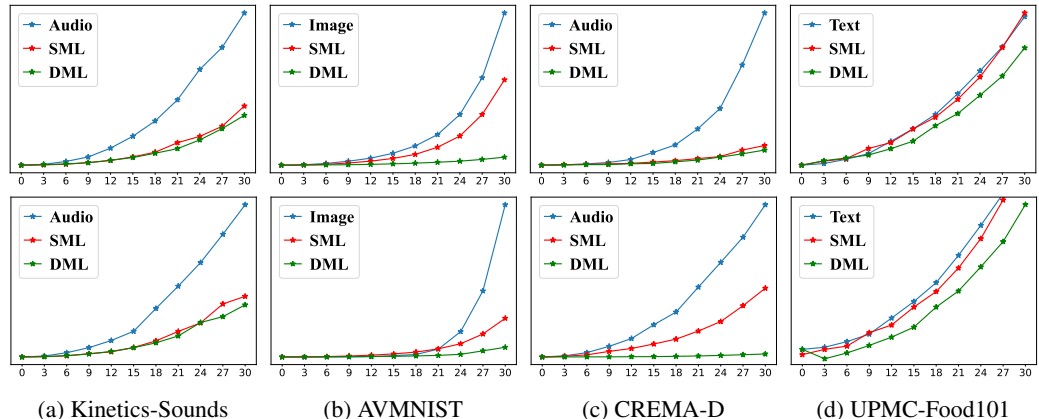

(a) Kinetics-Sounds      (b) AVMNIST      (c) CREMA-D      (d) UPMC-Food101

Figure 4: **Flatness Comparison between Modality.** These results shows that multimodal learning *generally yields improved flatness compared to single-modality training.* The top row reports evaluations on training datasets, whereas the bottom row reports evaluations on test datasets. In each plot, the $x$-axis denotes the perturbation radius and the $y$-axis represents $\mathcal{F}_{\text{gap}}(\theta)$.

a small random perturbation. A lower value of $\mathcal{F}_{\text{gap}}(\theta)$ indicates a flatter loss landscape of $\theta$. We sampled 100 random perturbations per model and perturbation radius. We then estimate $\mathcal{F}_{\text{gap}}$ via Monte Carlo approximation. We followed additional implementation settings proposed on SWAD.

As shown in Figure 4, models trained with multimodal data consistently exhibit lower perturbation sensitivity compared to unimodal learning, indicating a smoother and more stable optimization. Notably, our suggested DML framework shows the lowest $\mathcal{F}_{\text{gap}}$ tendency across settings, demonstrating flatter loss landscape. Importantly, this smoothing effect is observed not only on the training set but also on the test set, where DML approach leads to more robust model compare to SML. Thus, these results suggest that multimodal, especially when coupled with distributional alignment, enhances not only predictive accuracy but also robustness by inducing to smoother loss landscape.

## 5.2 MODALITY VARIANCE AND MODALITY GAP

Intrinsic to multimodal learning and as highlighted in Remark 2.2, modality gap and within-modality variance play a crucial role to reach to lower curvature of loss landscape. Building on this insight, we introduce two metrics: (1) modality variance, which captures the diversity of representations within each modality, and (2) modality gap that measures the divergence between modalities.

**Variance and Modality Gap in Multimodal Representations**  Directly comparing the input data across modalities is challenging due to the differing dimensionalities. We therefore analyze the latent vectors from each modality-specific encoder, which are projected into a same dimension of latent space. Although these representations may not fully capture the properties of original data, they could still serve as a reasonable proxy that reflects key characteristics of each modality (Liang et al., 2022).

To ensure fair comparisons across modalities, we normalize each latent representation vector $\mathbf{z}_i \in \mathbb{R}^d$ to have unit norm, i.e., $\|\mathbf{z}_i\|_2 = 1$. Given a set of $N$ such unit-norm vectors $\{\mathbf{z}_i\}_{i=1}^N$ from a particular modality, we compute the empirical covariance matrix: $\Sigma = \frac{1}{N} \sum_{n=1}^N (\mathbf{z}_n - \bar{\mathbf{z}})(\mathbf{z}_n - \bar{\mathbf{z}})^\top$, where $\bar{\mathbf{z}} = \frac{1}{N} \sum_{n=1}^N \mathbf{z}_n$. To summarize the variance of each modality, we use the trace $\text{Tr}(\Sigma)$, which captures the average directional spread. To quantify the *modality gap*, we compute the squared 2-Wasserstein distance between the empirical distributions $\mathcal{P}_1$ and $\mathcal{P}_2$ in the shared latent space:

$$W_2^2(\mathcal{P}_1, \mathcal{P}_2) = \inf_{\gamma \in \Gamma(\mathcal{P}_1, \mathcal{P}_2)} \int_{\mathcal{Z} \times \mathcal{Z}} \|\mathbf{z}_1 - \mathbf{z}_2\|_2^2 \, d\gamma(\mathbf{z}_1, \mathbf{z}_2),$$

where $\gamma(\mathcal{P}_1, \mathcal{P}_2)$ is the set of all joint couplings with marginals $\mathcal{P}_1$ and $\mathcal{P}_2$. We selected the Wasserstein distance for its ability to capture the beyond of geometric structure of given spaces and to provide a quantifiable measure of *how different two distributions are*—even in cases of minimal gap.

**Empirical Analysis: Effect of Variance and Modality Gap with t-SNE Visualization**  Building upon the theoretical framework introduced in Section 3, we analyzed how modality-specific char-

Table 5: **Modality variance and Wasserstein Distance (WD) across datasets.** Higher variance implies more diverse features within a modality (intra-modality), while higher WD indicates larger distributional discrepancy between modalities (inter-modality).

| Datasets | Kinetics-Sounds | | AVMNIST | | CREMA-D | | UPMC-Food101 | |
|---|---|---|---|---|---|---|---|---|
| | Audio | Visual | Image | Audio | Audio | Visual | Text | Image |
| **Variance**† (ratio) | 0.4803 (1.3465) | 0.3596 | 1.2773 (1.3162) | 0.9705 | 0.3782 (2.0270) | 0.1866 | 1.9269 (2.0602) | 0.9353 |
| **WD** | 0.5182 | | 0.5171 | | 0.5430 | | 0.5950 | |

†: Variances are scaled by $\times 10^3$.

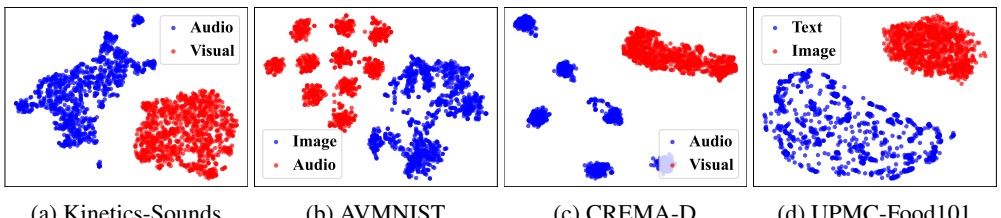

(a) Kinetics-Sounds      (b) AVMNIST      (c) CREMA-D      (d) UPMC-Food101

Figure 5: **Modality Gap Visualization**: We depict the modality gap via t-SNE visualization approach. The **blue scatter points** represents dominant modality data that contribute to higher performance, while the **red scatter points** correspond to the modality with lower performance.

acteristics—namely, variance and their modality gap to empirical performance. Table 5 reports the variance of each modality's latent representations and modality gap, Wasserstein Distance (WD). We compare given results from Table 5 with the empirical results in Table 1, 4, and Figure 4.

Based on our approaches, modalities with sufficient variance tend to produce more diverse latent representations when model parameters are near an optimal point. Moreover, with an adequate inter-modality distance (WD), the two modalities can interact more effectively and yield stronger multimodal benefits. This pattern is evident in Kinetics-Sounds and AVMNIST, where variance ratios and WD values are well balanced—showing neither pronounced skew nor excessive separation—leading to significant gains in multimodal learning, espeically in DML apporach, compared to unimodal baselines (Table 1; at least +5.32%) and substantial improvements in flatness metrics (Table 4). In contrast, CREMA-D and UPMC-Food101 still achieve improvement; yet resulting in marginal gains where these dataset exhibit unbalanced variance ratios dominated by one modality ($\times 2\uparrow$).

**t-SNE Visualization** To enhance understanding and interpretation, we provide t-SNE visualizations in Figure 5, which illustrated the intra- and inter-modality distributions for each dataset. These visualizations closely aligned the quantitative trends reported in Table 5, showing that datasets with more distinct intra- versus inter-modality characteristics are clearly separated in the latent space.

## 6 CONCLUSION

In this paper, we proposed the optimization process with theoretical analysis of multimodal learning through the lens of the loss landscape. Especially, our findings show that incorporating multiple modalities naturally leads to smoother regions of the loss surface, which we interpret as a form of convolutional smoothing induced by the additional modalities. Moreover, as inspired by this understanding, we proposed an empirical strategy that stochastically matches modality-paired samples with same given hypothesis space (e.g., label space), encouraging training from a more distributional perspective and yielding a smoother loss surface. Furthermore, we identify key future directions, like scalability, that are crucial for advancing practical application of multimodal learning based on our theoretical understanding.

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

## A  MATHEMATICAL PROOFS

We first recall our notations, assumptions, and the definition:

**Notations:**

- $\mathbf{x}_i \in \mathcal{X}_i \subset \mathbb{R}^{d_i}, \mathbf{x}_j \in \mathcal{X}_j \subset \mathbb{R}^{d_j}$ denote input data from two different modalities $i$ and $j$, with their data distribution $\mathbf{x}_i \sim p(\mathbf{x}_i)$ and $\mathbf{x}_j \sim p(\mathbf{x}_j)$, respectively.

- $y \in \mathcal{Y}$ be the target label, which indicates the shared semantics across modalities.

- $f_i(\mathbf{x}_i), f_j(\mathbf{x}_j) \in \mathbb{R}^d$ are the modality encoders for the respective modality.

- $f(\mathbf{x}_i, \mathbf{x}_j; \theta) := \phi(f_i(\mathbf{x}_i), f_j(\mathbf{x}_j))$ be a function parametrized by $\theta \in \mathbb{R}^m$, with modality specific function $f_i(\cdot)$ and $f_j(\cdot)$, and modality fusion function $\phi(\cdot, \cdot)$

- $\ell(f(\mathbf{x}_i, \mathbf{x}_j; \theta), y)$ is the loss function.

- Eq. 1: $\mathcal{L}_{\text{uni}} = \underset{y \in \mathcal{Y}}{\mathbb{E}} \left[ \int \ell(f(\mathbf{x}_i, \bar{\mathbf{x}}_j; \theta), y) p(\mathbf{x}_i \mid y) \, d\mathbf{x}_i \right]$

- Eq. 2: $\mathcal{L}_{\text{multi}} = \underset{y \in \mathcal{Y}}{\mathbb{E}} \left[ \iint \ell(f(\mathbf{x}_i, \mathbf{x}_j; \theta), y) p(\mathbf{x}_i, \mathbf{x}_j \mid y) \, d\mathbf{x}_i \, d\mathbf{x}_j \right]$

**Definition 1.** (**Approximate Translation-Invariance (ATI)**) *Let $\phi : \mathbb{R}^{d_i} \times \mathbb{R}^{d_j} \to \mathbb{R}^d$ be a mapping function. Then, we define that $\phi$ exhibits Approximate Translation-Invariance (ATI) if there exists a vector $\boldsymbol{\alpha} \in \mathbb{R}^d$ such that*

$$\phi(\mathbf{u}, \mathbf{v} + \boldsymbol{\tau}) = \phi(\mathbf{u}, \mathbf{v}) + \boldsymbol{\alpha} \odot \boldsymbol{\tau}$$

*for all $\mathbf{u}, \mathbf{v}, \boldsymbol{\tau} \in \mathbb{R}^d$, where $\odot$ denotes element-wise (Hadamard) product.*

**Assumption 1.** *Let $\phi$ be a modality fusion function. Then, we assume that $\phi$ satisfies the Approximate Translation-Invariance (ATI) property.*

**Assumption 2.** (**Conditional Independence**) *With given $y$, the two modalities are conditionally independent: $p(\mathbf{x}_i, \mathbf{x}_j \mid y) = p(\mathbf{x}_i \mid y) p(\mathbf{x}_j \mid y)$.*

**Definition 2.**(**Fusion-Induced Scaled Shift and Kernel**) *Let $\bar{\mathbf{x}}_j$ denote input data representing absence of modality $j$, such as zero vector or constant value, and define fusion-induced scaled shift as :*

$$\tau_{\boldsymbol{\alpha}}(\mathbf{x}_j) := \boldsymbol{\alpha} \odot \left( \phi(f_i(\mathbf{x}_i), f_j(\mathbf{x}_j)) - \phi(f_i(\mathbf{x}_i), f_j(\bar{\mathbf{x}}_j)) \right),$$

*The corresponding* scaled-shift kernel *is then defined using the Dirac delta function $\delta_{\tau_{\boldsymbol{\alpha}}(\mathbf{x}_j)}(\cdot)$ centered at $\tau_{\boldsymbol{\alpha}}(\mathbf{x}_j)$:*

$$\mathcal{K}_{\mathbf{x}_j, \boldsymbol{\alpha}}(\boldsymbol{\tau}) := \underset{\mathbf{x}_j \sim p(\mathbf{x}_j \mid y)}{\mathbb{E}} \left[ \delta_{\tau_{\boldsymbol{\alpha}}(\mathbf{x}_j)}(\boldsymbol{\tau}) \right],$$

### A.1  PROOFS OF THEOREM 1

Before the proof, we recall our Theorem 1:

**Theorem 1.** (**Convolutional Smoothing with Modality Scaled Shifted Kernel**) *Under the Assumption 1, the expected multimodal loss is a scaled convolution of the unimodal loss:*

$$\mathcal{L}_{\text{multi}} = \underset{y \in \mathcal{Y}}{\mathbb{E}} \left[ \left( \mathcal{L}_{\text{uni}} \circledast \mathcal{K}_{\mathbf{x}_j, \boldsymbol{\alpha}} \right) (f \mid y) \right].$$

*Proof.* According to Eq. 2, we express the multimodal loss in terms of $\tau_{\boldsymbol{\alpha}}(\mathbf{x}_j)$ from Definition 2:

$$
\begin{aligned}
\mathcal{L}_{\text{multi}}(\cdot \mid y) &= \iint \ell\big(\phi(f_i(\mathbf{x}_i), f_j(\mathbf{x}_j)), y\big) \, p(\mathbf{x}_i, \mathbf{x}_j \mid y) \, d\mathbf{x}_i \, d\mathbf{x}_j \\
&= \iint \ell\big(\phi(f_i(\mathbf{x}_i), f_j(\bar{\mathbf{x}}_j)) + \tau_{\boldsymbol{\alpha}}(\mathbf{x}_j), y\big) \, p(\mathbf{x}_i, \mathbf{x}_j \mid y) \, d\mathbf{x}_i \, d\mathbf{x}_j \\
&= \iint \ell\big(\phi(f_i(\mathbf{x}_i), f_j(\bar{\mathbf{x}}_j)) + \tau_{\boldsymbol{\alpha}}(\mathbf{x}_j), y\big) \, p(\mathbf{x}_i \mid y) \, p(\mathbf{x}_j \mid y) \, d\mathbf{x}_i \, d\mathbf{x}_j,
\end{aligned}
$$

Let $f := \phi(f_i(\mathbf{x}_i), f_j(\bar{\mathbf{x}}_j)) \in \mathbb{R}^d$. Then the above becomes

$$\mathcal{L}_{\text{multi}}(\cdot \mid y) = \int \left[ \int \ell\big(f + \tau_{\boldsymbol{\alpha}}(\mathbf{x}_j), y\big) \, p(\mathbf{x}_j \mid y) \, d\mathbf{x}_j \right] p(\mathbf{x}_i \mid y) \, d\mathbf{x}_i.$$

By Definition 2, we set the scaled-shift kernel

$$\mathcal{K}_{\mathbf{x}_j, \boldsymbol{\alpha}}(\boldsymbol{\tau}) := \mathbb{E}_{\mathbf{x}_j \sim p(\mathbf{x}_j \mid y)} \big[ \delta_{\tau_{\boldsymbol{\alpha}}(\mathbf{x}_j)}(\boldsymbol{\tau}) \big],$$

where $\delta_{\tau_{\boldsymbol{\alpha}}(\mathbf{x}_j)}$ is the Dirac delta centered at $\tau_{\boldsymbol{\alpha}}(\mathbf{x}_j)$. The inner integral over $\mathbf{x}_j$ is therefore the integral of $\ell(f + \boldsymbol{\tau}, y)$ against this kernel, i.e.

$$\int \ell(f + \boldsymbol{\tau}, y) \, \mathcal{K}_{\mathbf{x}_j, \boldsymbol{\alpha}}(\boldsymbol{\tau}) \, d\boldsymbol{\tau} = \big(\ell(\cdot, y) \circledast \mathcal{K}_{\mathbf{x}_j, \boldsymbol{\alpha}}\big)(f).$$

Hence the conditional multimodal loss is the convolution of the unimodal loss with the scaled-shift kernel. Taking expectation over $y \in \mathcal{Y}$ completes the proof:

$$\mathcal{L}_{\text{multi}} = \mathbb{E}_{y \in \mathcal{Y}} \big[ \big(\mathcal{L}_{\text{uni}} \circledast \mathcal{K}_{\mathbf{x}_j, \boldsymbol{\alpha}}\big) (f \mid y) \big].$$

$\square$

**Theoretical Justification of $\mathbf{x}_j$.** The smoothing effect described in Theorem 1 critically depends on the auxiliary modality $\mathbf{x}_j$ containing label-dependent information. If $\mathbf{x}_j$ is statistically independent of the label (i.e., $p(\mathbf{x}_j \mid y) = p(\mathbf{x}_j)$), then the optimal training objective will suppress its influence.

Consider, for example, the degenerate case where $\mathbf{x}_j \sim \mathcal{N}(0, \sigma^2 I)$ is high-variance Gaussian noise independent of $y$. In this case, for each $\mathbf{x}_i$, the model minimizes

$$\min_{\theta} \mathbb{E}_{\mathbf{x}_j \sim p(\mathbf{x}_j)} \big[ \ell(f(\mathbf{x}_i, \mathbf{x}_j; \theta), y) \big],$$

and because $\mathbf{x}_j$ carries no label information, the optimal mapping tends to a constant representation $f_j(\mathbf{x}_j) \approx \mathbf{c}$ for some fixed vector $\mathbf{c}$. The fusion-induced shift (Definition 2) then becomes approximately deterministic, producing an induced kernel that collapses to a Dirac delta at the origin:

$$\mathcal{K}_{\mathbf{x}_j, \boldsymbol{\alpha}} = \delta_0(\boldsymbol{\tau}).$$

According to Theorem 1, convolution with a Dirac delta kernel leaves the loss unchanged:

$$\mathcal{L}_{\text{multi}} = \mathcal{L}_{\text{uni}} \circledast \delta_0 = \mathcal{L}_{\text{uni}}.$$

Thus, even though the input noise of $\mathbf{x}_j$ has high variance, the model's training suppresses it entirely, leading to zero variance in the shift kernel and no smoothing effect. This analysis reinforces that meaningful convolutional smoothing—and thus our theoretical results—require the auxiliary modality to be label-dependent. When $\mathbf{x}_j$ is non-informative, the multimodal loss naturally collapses to the unimodal form, confirming the role of label dependence in generating useful smoothing behavior.

## A.2 PROOFS OF THEOREM 2

We also recall our Theorem 2 before the proof:

**Theorem 2 (Smooth Loss Landscape of Multimodal Learning)** *Suppose that conditional unimodal loss $\mathcal{L}_{uni}(f \mid y)$ is continuous and bounded for all $y \in \mathcal{Y}$, and the shift kernel $\mathcal{K}_{\mathbf{x}_j, \boldsymbol{\alpha}}$ has finite variance. Then the $\mathcal{L}_{multi}$ is smoother than the $\mathcal{L}_{uni}$, in the following two folds:*

1. ***Spectral Norm Upper Bound on Hessian:*** *For all $f$, the largest spectral norm of the multimodal Hessian is upper-bounded by that of the unimodal loss:*

$$\sup_f \|\nabla^2 \mathcal{L}_{multi}(f \mid y)\|_2 \leq \sup_f \|\nabla^2 \mathcal{L}_{uni}(f \mid y)\|_2.$$

2. ***Frequency Domain Interpretation:*** *Let $\widehat{\mathcal{L}}_{multi}(\omega)$, $\widehat{\mathcal{L}}_{uni}(\omega)$, and $\widehat{\mathcal{K}}_{\mathbf{x}_j, \boldsymbol{\alpha}}(\omega)$ denote the Fourier transforms of $\mathcal{L}_{multi}$, $\mathcal{L}_{uni}$, and $\mathcal{K}_{\mathbf{x}_j, \boldsymbol{\alpha}}$ respectively. Then, Eq. 5 can be expressed in the frequency domain as: $\widehat{\mathcal{L}}_{multi}(\omega) = \widehat{\mathcal{L}}_{uni}(\omega) \cdot \widehat{\mathcal{K}}_{\mathbf{x}_j, \boldsymbol{\alpha}}(\omega)$. Moreover, it follows that:*

$$\mathcal{O}(\widehat{\mathcal{L}}_{multi}(\omega)) \leq \mathcal{O}(\widehat{\mathcal{L}}_{uni}(\omega)).$$

*Proof.* We prove each of the statements in the Theorem 2 as follows:

**1. Spectral Norm Upper Bound on the Multimodal Hessian**    We prove that the maximal spectral norm of the Hessian of the multimodal loss is upper-bounded by that of the unimodal loss.

Let the Hessians of the unimodal and multimodal losses at a point $f \in \mathbb{R}^d$ be defined as:

$$H_{\text{uni}}(f \mid y) := \nabla^2 \mathcal{L}_{\text{uni}}(f \mid y), \qquad H_{\text{multi}}(f \mid y) := \nabla^2 \mathcal{L}_{\text{multi}}(f \mid y).$$

From the convolutional smoothing formulation (Eq. 5), the multimodal loss is given by:

$$\mathcal{L}_{\text{multi}}(f \mid y) = \left(\mathcal{L}_{\text{uni}}(\cdot \mid y) \circledast \mathcal{K}_{\mathbf{x}_j, \boldsymbol{\alpha}}\right)(f) = \int \mathcal{L}_{\text{uni}}(f + \boldsymbol{\tau} \mid y)\, \mathcal{K}_{\mathbf{x}_j, \boldsymbol{\alpha}}(\boldsymbol{\tau})\, d\boldsymbol{\tau},$$

so differentiating twice with respect to $f$ yields

$$H_{\text{multi}}(f \mid y) = \int H_{\text{uni}}(f + \boldsymbol{\tau} \mid y)\, \mathcal{K}_{\mathbf{x}_j, \boldsymbol{\alpha}}(\boldsymbol{\tau})\, d\boldsymbol{\tau}.$$

Taking the spectral norm on both sides yields:

$$\|H_{\text{multi}}(f \mid y)\|_2 = \left\| \int H_{\text{uni}}(f + \boldsymbol{\tau} \mid y)\, \mathcal{K}_{\mathbf{x}_j, \boldsymbol{\alpha}}(\boldsymbol{\tau})\, d\boldsymbol{\tau} \right\|_2$$

$$\leq \int \left\| H_{\text{uni}}(f + \boldsymbol{\tau} \mid y) \right\|_2 \mathcal{K}_{\mathbf{x}_j, \boldsymbol{\alpha}}(\boldsymbol{\tau})\, d\boldsymbol{\tau}.$$

where the inequality follows from Jensen's inequality, given that the spectral norm $\| \cdot \|_2$ is a convex function over symmetric matrices. Now, we take the supremum over $f \in \mathbb{R}^d$. For every fixed $\boldsymbol{\tau}$ the map $f \mapsto f + \boldsymbol{\tau}$ is a bijection of $\mathbb{R}^d$ with additional notation $u = f + \boldsymbol{\tau}$, hence

$$\sup_f \left\| H_{\text{uni}}(f + \boldsymbol{\tau} \mid y) \right\|_2 = \sup_u \left\| H_{\text{uni}}(u \mid y) \right\|_2,$$

Then, we derive the bound of the largest spectral norm of the Hessian of multimodal learning relative to unimodal learning as follows:

$$\sup_f \|H_{\text{multi}}(f \mid y)\|_2 \leq \sup_f \int \left\| H_{\text{uni}}(f + \boldsymbol{\tau} \mid y) \right\|_2 \mathcal{K}_{\mathbf{x}_j, \boldsymbol{\alpha}}(\boldsymbol{\tau})\, d\boldsymbol{\tau}$$

$$\leq \int \left( \sup_u \left\| H_{\text{uni}}(u \mid y) \right\|_2 \right) \mathcal{K}_{\mathbf{x}_j, \boldsymbol{\alpha}}(\boldsymbol{\tau})\, d\boldsymbol{\tau}$$

$$= \sup_u \left\| H_{\text{uni}}(u \mid y) \right\|_2 \cdot \int \mathcal{K}_{\mathbf{x}_j, \boldsymbol{\alpha}}(\boldsymbol{\tau})\, d\boldsymbol{\tau}$$

$$= \sup_f \left\| H_{\text{uni}}(f \mid y) \right\|_2$$

The final equality holds for two reasons. First, $\mathcal{K}_{\mathbf{x}_j, \boldsymbol{\alpha}}$ is a probability density function, and this satisfies $\int \mathcal{K}_{\mathbf{x}_j, \boldsymbol{\alpha}}(\boldsymbol{\tau})\, d\boldsymbol{\tau} = 1$. Second, because the supremum is taken over all of $\mathbb{R}^d$, the variable is a dummy variable, and this means it is valid to rename $u \mapsto f$.

Finally, the supremum of the Hessian of the multimodal loss is bounded by that of the unimodal loss,

$$\sup_f \left\| \nabla^2 \mathcal{L}_{\text{multi}}(f \mid y) \right\|_2 \leq \sup_f \left\| \nabla^2 \mathcal{L}_{\text{uni}}(f \mid y) \right\|_2,$$

This result implies that $\mathcal{L}_{\text{multi}}$ landscape, in terms of its maximal curvature (i.e., the largest spectral norm of the Hessian), is globally smoother or at most equally sharp as that of the $\mathcal{L}_{\text{uni}}$.

**2. Frequency Domain Interpretation**    Consider loss function $\mathcal{L}(f(\cdot, \cdot; \theta))$, where $f$ is parameterized by $\theta \in \mathbb{R}^m$. We treat loss as a function over the parameter space $\theta$, and analyze its behavior in the frequency domain. We define the Fourier transform of the loss function with respect to $\theta$ as:

$$\widehat{\mathcal{L}}(\omega) = \int \mathcal{L}(f(\cdot, \cdot; \theta))\, e^{-i\omega^\top \theta}\, d\theta, \quad \omega \in \mathbb{R}^m.$$

By the convolution theorem in Fourier analysis, the Fourier transform of the convolution of two functions equals the pointwise product of their individual Fourier transforms:

$$\mathcal{F}[f \circledast g](\omega) = \mathcal{F}[f](\omega) \cdot \mathcal{F}[g](\omega).$$

Applying this to Eq. 5, which expresses the $\mathcal{L}_{\text{multi}}$ as a convolution over parameter space, we obtain:

$$\widehat{\mathcal{L}}_{\text{multi}}(\omega) = \widehat{\mathcal{L}}_{\text{uni}}(\omega) \cdot \widehat{\mathcal{K}}_{\mathbf{x}_j, \boldsymbol{\alpha}}(\omega),$$

where $\widehat{\mathcal{K}}_{\mathbf{x}_j, \boldsymbol{\alpha}}(\omega)$ denotes the Fourier transform of the shift kernel applied in the parameter space $\theta$.

To analyze the asymptotic behavior, we consider the decay properties of $\widehat{\mathcal{L}}_{\text{multi}}(\omega)$ and $\widehat{\mathcal{L}}_{\text{uni}}(\omega)$ as $\|\omega\| \to \infty$. Moreover, this consideration shows that the decay rate of $\widehat{\mathcal{K}}_{\mathbf{x}_j, \boldsymbol{\alpha}}(\omega)$ decides how much high-frequency content is preserved or attenuated by the $\mathcal{L}_{\text{multi}}$. Since the shift kernel is defined as: $\mathcal{K}_{\mathbf{x}_j, \boldsymbol{\alpha}}(\boldsymbol{\tau}) := \mathbb{E}_{\mathbf{x}_j \sim p(\mathbf{x}_j|y)} \left[ \delta_{\tau_{\boldsymbol{\alpha}}(\mathbf{x}_j)}(\boldsymbol{\tau}) \right]$, which is the expectation of a Dirac delta centered at $\tau_{\boldsymbol{\alpha}}(\mathbf{x}_j)$, then its Fourier transform is the *characteristic function* of the random variable $\tau_{\boldsymbol{\alpha}}(\mathbf{x}_j)$ as follows:

$$\widehat{\mathcal{K}}_{\mathbf{x}_j, \boldsymbol{\alpha}}(\omega) = \mathbb{E}_{\mathbf{x}_j \sim p(\mathbf{x}_j|y)} \left[ e^{-i\omega^\top \tau_{\boldsymbol{\alpha}}(\mathbf{x}_j)} \right].$$

As we assumed that $\tau_{\boldsymbol{\alpha}}(\mathbf{x}_j)$ has finite variance, this characteristic function is continuous and satisfies the following properties (Körner, 1988; Billingsley, 1995)::

$$\left| \widehat{\mathcal{K}}_{\mathbf{x}_j, \boldsymbol{\alpha}}(\omega) \right| \leq 1, \quad \text{and} \quad \lim_{\|\omega\| \to \infty} \left| \widehat{\mathcal{K}}_{\mathbf{x}_j, \boldsymbol{\alpha}}(\omega) \right| = 0.$$

Consequently, the magnitude of $\mathcal{L}_{\text{multi}}$ in the frequency domain is bounded above by that of the $\mathcal{L}_{\text{uni}}$:

$$\left| \widehat{\mathcal{L}}_{\text{multi}}(\omega) \right| = \left| \widehat{\mathcal{L}}_{\text{uni}}(\omega) \right| \cdot \left| \widehat{\mathcal{K}}_{\mathbf{x}_j, \boldsymbol{\alpha}}(\omega) \right| \leq \left| \widehat{\mathcal{L}}_{\text{uni}}(\omega) \right|$$

This implies that the high-frequency components of the $\mathcal{L}_{\text{multi}}$ are effectively suppressed relative to those of the $\mathcal{L}_{\text{uni}}$. Hence, in terms of asymptotic decay:

$$\mathcal{O} \left( \widehat{\mathcal{L}}_{\text{multi}}(\omega) \right) \leq \mathcal{O} \left( \widehat{\mathcal{L}}_{\text{uni}}(\omega) \right).$$

This frequency-domain analysis highlights the smoothing effect induced by the $\mathcal{L}_{\text{multi}}$ formulation, which naturally dampens high-frequency variations in the parameter space $\theta \in \mathbb{R}^m$. $\qquad\square$

# B  EXPERIMENTAL DETAILS

## B.1  MULTIMODAL DATASETS DESCRIPTIONS

**Kinetics-Sounds** Kinetics-Sounds (Arandjelović & Zisserman, 2018) is a multimodal dataset that comprises both audio and visual modalities, derived as a subset of the Kinetics-400 dataset (Kay et al., 2017). The Kinetics-Sounds contains 34 human action classes with . Each clip has an approximate duration of 10 seconds and is sourced from distinct YouTube videos. For the visual modality preprocessing, we extract video frames at a rate of 1 frame per second, with each frame resized to $224 \times 224 \times 3$. During iterative training, we employ a random sampling strategy where 3 frames are selected per class, resulting in a total sampling of batch size $\times$ 1. The audio modality contains sound clips from the same videos, which are converted into mel-spectrograms with a resolution of $128 \times 128 \times 1$. We split training set with a 9:1 ratio to create validation set.

**AVMNIST** AVMNIST (Vielzeuf et al., 2018) is a multimodal dataset comprising audio and visual modalities. The visual modality consists of digit images (0–9) from MNIST (Lecun et al., 1998), with each image having a resolution of $28 \times 28 \times 1$. These images is PCA-projected, retaining 75% of the energy. The audio modality includes spoken digit recordings from the Free Spoken Digit Dataset (FSDD) (Jackson et al., 2018). The audio samples are preprocessed into mel-spectrograms, each with a resolution of $112 \times 112 \times 1$. We split training datasets into 9:1 ratio to create validation datasets.

**CREMA-D** CREMA-D (Cao et al., 2014) is an audio-visual modality dataset designed for speech emotion recognition, consisting of 7,442 video clips from 91 actors (48 male and 43 female). The dataset includes six emotions: *Anger, Disgust, Fear, Happy, Neutral, and Sad.* For the visual modality, we pre-process 1 frame per second of each video. During training, we randomly sample one frame per class, resulting in an overall image batch size of (batch size $\times$ 1). Audio data is preprocessed by resampling into 22,050 Hz and converted into spectrograms using the Short-Time Fourier Transform (STFT) with a 512-point FFT and a hop length of 353, and then log-scale magnitude of spectrograms. Following prior works (Peng et al., 2022; Li et al., 2023), we adopt the same dataset splitting strategy. As the original dataset provides only training and test datasets, we further randomly split the training dataset into a 9:1 ratio to create a validation split.

**UPMC-Food101**    UPMC-Food101 (Wang et al., 2015) is a large-scale multimodal dataset comprising paired text and image modalities, specifically curated for food recognition and recipe understanding tasks. The dataset encompasses 101 different food categories with corresponding recipes, with over 100,000 items collected from the web. For preprocessing the visual modality, all food images are standardized by resizing to $224 \times 224 \times 3$. For the text modality, which contains recipe descriptions and ingredient lists, we use the BERT model (Devlin et al., 2018) along with its corresponding tokenizer, a widely adopted choice for encoder-based transformer architectures in recent studies. Since a validation set is not provided, we split the original training set using a 9:1 ratio to create one.

## B.2    SETUPS

**Computational Resources**    We utilized the configuration of computational resources and frameworks as follows:

- **CPU**: Intel(R) Xeon(R) Gold 6342 @ 2.80GHz with 256GB RAM
- **GPU**: Single NVIDIA RTX A5000 with 24GB VRAM
- **Deep Learning Framework**: PyTorch 2.0.1 with CUDA 11.8
- **Codebase**: Adopted official implementation of Adaptive Gradient Modulation (Li et al., 2023)

**Hyperparameter Details**    We provide experimental details for all datasets in Table 6. We mostly modified the configuration that has been provided in BalanceBenchmark (Xu et al., 2025).

Table 6: **Hyperparameter Settings**

| Settings | Kinetics-Sounds | AVMNIST | CREMA-D | UPMC-Food101 |
|---|---|---|---|---|
| Epochs | 70 | 70 | 70 | 70 |
| Batch Size | 64 | 64 | 16 | 64 |
| Learning Rate (LR) | 0.001 | 0.001 | 0.001 | 0.001 (text) / 0.01 (image) |
| Weight Decay | 0.0001 | 0.0001 | 0.0005 | 0.001 (text) / 0.0001 (image) |
| Optimizer | Adam | SGD | SGD | SGD |
| Scheduler | StepLR | StepLR | StepLR | StepLR |
| Decay Step / Ratio | 30 / 0.1 | 30 / 0.1 | 50 / 0.1 | 30 / 0.1 |
| Model Architecture | ResNet | ResNet | ResNet | (Pretrain) BERT & ResNet-18 |

## C    ADDITIONAL EXPERIMENTS AND ANALYSES

### C.1    APPLICABILITY OF DML IN VARIOUS TASKS

While our suggested approach, DML, has been primarily evaluated on supervised classification tasks, its core principle—leveraging shared semantic or feature-level information conditioned on label-like signals—is not limited to supervised settings. In fact, DML can be extended to self-supervised frameworks, such as contrastive learning (e.g., CLIP), where pairing can be guided by pseudo-labels or semantic similarity rather than ground-truth annotations.

To validate this, we applied DML to a contrastive learning setup using the **Flickr-30k** dataset (Plummer et al., 2015). We fine-tuned a model initialized from the pretrained CLIP-B/16 (Radford et al., 2021). Then, since Flickr-30k does not provide the explicit class annotations required by DML, we constructed pseudo-labels through the following procedure. We first extracted image features using a pretrained ViT-L/14 model , then performed dimensionality reduction with UMAP (McInnes et al., 2018). Then, we applied $K$-means clustering with $K = 100$ to partition the embedding space. Each image was assigned to the nearest cluster centroid, which we treated as its pseudo-class label.

This pseudo-labeling scheme allowed us to construct class-aware pairs for DML, enabling evaluation in scenarios where ground-truth labels are unavailable. We trained the model using the InfoNCE loss (Oord et al., 2018; Radford et al., 2021) and evaluated performance on both **Image-to-Text (I2T)** and **Text-to-Image (T2I)** retrieval tasks.

The retrieval results in Table 7 show consistent improvements across most metrics. For instance, the SML approach achieves an R@1 score of 87.34% and an R@5 score of 97.10% for I2T retrieval.

Table 7: Retrieval performance of SML (fixed pairing) and DML (shuffled pairing) on Flickr-30k. Metrics are reported as Recall@K (%).

| Tasks | Image-to-Text (I2T) | | | Text-to-Image (T2I) | | |
|---|---|---|---|---|---|---|
| Metrics | R@1 | R@5 | R@10 | R@1 | R@5 | R@10 |
| SML (Fixed Pair) | 87.34 | 97.10 | 99.56 | 74.19 | 89.77 | 93.53 |
| DML (Shuffled Pair) | 89.10 | 97.56 | 99.31 | 76.10 | 90.51 | 93.89 |

Applying DML yields a targeted improvement, increasing R@1 to 89.10% (**+1.76%** over SML) and R@5 to 97.56% (**+0.46%** over SML). For R@10, SML slightly outperforms DML, although the overall scores remain high. Similar trends are observed in T2I retrieval, where SML attains 74.19% at R@1 and 89.77% at R@5. DML further improves R@1 to 76.10% (**+1.91%** over SML) and 90.51% (**+0.74%** over SML) at R@5, with additional gains and R@10.

These results consistently show that distributionally matched supervision (DML) outperforms supervision from exactly matched pairs (SML). This demonstrates that class-level stochastic alignment is an effective method for improving retrieval performance, even without explicit pair-level supervision.

## C.2 COMPLEMENTARY EFFECTS OF DML AND DATA AUGMENTATION (DA)

As mentioned on Section 4.3, given that DML introduces stochasticity during training, one might intuitively interpret it as a form of Data Augmentation (DA). Consequently, it is natural to question whether the improvements in accuracy and loss landscape smoothness are merely artifacts of standard augmentation effects rather than a distinct theoretical mechanism. Therefore, in this section, we clarify the relationship between DML and conventional DA. We show that while they share high-level goals, DML and DA is fundamentally different principles, and even complementary approaches.

**Input Space (DA) vs. Hypothesis Space (DML)** First, we would like to emphasize the difference between DML and traditional DA. Conventional DA operates on the input space, applying semantics-preserving perturbations (e.g., cropping, rotation, jitter for visual data), where its goal is to expand the distribution within the original semantic boundaries. DML, however, actively constructs such novel pairings. DML generates pairs by matching instances that share the same hypothesis but originate from different data points.

For example, consider two pairs: `[Poodle Image, Poodle Audio]` and `[Bichon Image, Bichon Audio]`, both labeled as "dog." Traditional DA can only perturb these pairs individually, but cannot generate a novel pair such as `[Poodle Image, Bichon Audio]`. DML, however, can create such a novel pair by leveraging shared labels. This inter-modality stochasticity encourages the model to learn more diverse semantics. Moreover, DML is complementary to conventional DA. One could first generate `[Poodle Image, Bichon Audio]` via DML, and then apply standard DA (e.g., cropping, jittering) to this new pair. This demonstrates that DML introduces a mechanism fundamentally distinct from, yet compatible with, traditional DA.

**Results** To empirically validate the complementarity of our approach, we applied standard data augmentation (DA) techniques to the image modality while training both SML and DML. In Table 8, **J** represents Color **J**itter, **C** denotes **C**ropping, and **F** denotes **F**lipping. For color jitter, we used standard parameters: brightness = 0.4, contrast = 0.4, saturation = 0.4, hue = 0.1, and a flipping probability of 0.5.

As shown in Table 8, applying DA alone yields modest improvements for both SML and DML. In contrast, combining DML with DA consistently produces higher performance across all datasets. Specifically, DML is better able to leverage the increased data diversity compared to the baseline. For instance, on the UPMC-Food101 dataset, combining all three augmentations (J+C+F) raises DML's accuracy to 94.93% (**+2.21%**), significantly outperforming SML with all augmentations, which achieves 92.72%. Similar trends are observed on other datasets, where DML with all augmentations improves performance by at least +1.12% and up to +2.68%.

More importantly, these results and recent finding support Theorem 2, which suggest that increasing data variance leads to better model performance in multimodal learning: the performance consistently

improves as additional DA techniques are incorporated. Notably, DML combined with all three augmentations achieves the highest performance across all benchmarks, confirming the validity of our hypothesis and demonstrating the complementary effect of DA in this context.

Table 8: **Effect of Data Augmentation (DA) on Multimodal Learning**

| Data Augmentation | | | KS | AVMNIST | CREMA-D | UPMC-Food101 |
|---|---|---|---|---|---|---|
| SML | 1 Aug. | Original | 62.95 | 70.35 | 59.62 | 91.48 |
| | | + J | 63.87 | 70.39 | 59.66 | 91.95 |
| | | + C | 63.19 | 70.59 | 59.10 | 92.13 |
| | | + F | 64.14 | 70.87 | 59.35 | 91.99 |
| | 2 Aug. | + J + C | 63.89 | 70.97 | 59.95 | 92.08 |
| | | + J + F | 64.18 | 70.94 | 61.02 | 92.55 |
| | | + C + F | 63.78 | 70.37 | 60.83 | 91.93 |
| | 3 Aug. | + J + C + F | 64.51 | 71.89 | 60.83 | 92.72 |
| DML | 1 Aug. | Original | 65.89 | 71.69 | 59.77 | 92.84 |
| | | + J | 66.49 | 71.22 | 59.94 | 93.57 |
| | | + C | 65.77 | 71.80 | 60.57 | 93.21 |
| | | + F | 66.85 | 72.07 | 60.51 | 94.29 |
| | 2 Aug. | + J + C | 65.68 | 71.95 | 61.69 | 93.72 |
| | | + J + F | 66.15 | 72.47 | 61.84 | 94.49 |
| | | + C + F | 66.64 | 72.14 | 61.56 | 93.86 |
| | 3 Aug. | + J + C + F | 67.19 | 73.01 | 61.99 | 94.93 |

**Smoothing Perspective**   Even if DML is interpreted broadly as a form of DA, this perspective reinforces the theoretical novelty of our approach. Recent work (Yoo & Yoon, 2025) demonstrates that DA not only increases data quantity, but also induces a smoothing effect: by introducing perturbations in the parameter space, DA guides optimization toward flatter minima, leading to tighter generalization bounds. Analogously, DML smooths the loss landscape through stochastic cross-modal pairings. By effectively exploring the hypothesis space, DML promotes robustness and generalizability in a manner comparable to the smoothing effect of conventional DA, but in a fundamentally different dimension—the label or hypothesis space rather than the input space. Therefore, while DML aligns with DA in the goal of generalization, it is differentiated by its cross-modal pairing mechanism in the hypothesis space. This approach complements input-space DA and provides a theoretical justification for DML's effectiveness, particularly regarding its smoothing effect on the loss landscape.

## C.3   ADDITIONAL COMPARISON WITH FLATNESS-AWARE APPROACHES

Here, we conduct additional experiments to compare DML with recent flatness-aware approaches. Table 9 reports results on multimodal datasets, benchmarking DML against Label Smoothing (LS), Stochastic Weight Averaging (SWA), and Sharpness-Aware Minimization (SAM), each applied directly to our SML, where all results are averaged over 3 times. These conjunctions with SML can be understood as the baselines of multimodal training of LS, SWA, and SAM.

As shown in Table 9, DML generally outperforms the compared flatness-aware approaches, with CREMA-D being the only minor exception. Across these three multimodal datasets, the improvements range from approximately **+1.1%** to **+1.7%**, reflecting consistent gains beyond what is achieved by LS, SWA, or SAM. These results highlight that DML not only offers a straightforward alternative to conventional flatness-aware techniques but also delivers reliably stronger empirical performance across diverse multimodal settings.

Table 9: **Performance Comparison of Flatness Approaches and DML**

| Methods | KS | AVMNIST | CREMA-D | UPMC Food101 |
|---------|-----|---------|---------|--------------|
| SML | 62.95 | 70.30 | 59.62 | 91.48 |
| SML + LS | 63.03 | 70.44 | 59.53 | 91.79 |
| SML + SWA | 64.88 | 69.24 | 59.01 | 92.08 |
| SML + SAM | 64.19 | 70.91 | **60.51** | 91.76 |
| DML | **65.89** | **71.69** | 59.77 | **92.84** |

## C.4 EXTENDING DML TO SETTINGS WITH 3 MODALITIES

In this section, we present an extensive evaluation of our method on a 3-modality dataset using the UR-Funny benchmark. This dataset incorporates text, visual, and audio information and is specifically designed for humor prediction tasks. Humor is expressed through verbal content (text), gestures and facial expressions (visual), and prosodic cues (audio), with all samples collected from TED Talks. For architectural choices and hyperparameters, we generally follow the settings used for the CREMA-D experiments, with the exception that we adopt a transformer-based model following the design in Hasan et al. (2019).

We report the experimental results in Table 10, covering classification accuracy, the Hessian maximum eigenvalue ($\lambda_{\max}$) (Ghorbani et al., 2019), and the LPF metric ($\mathcal{L}_{\mathrm{LPF}}$) (Bisla et al., 2022) as described in Section 5.1. As shown in the results, DML improves accuracy by a clear margin (+1.21%) and also achieves flatter minima, as reflected in both the reduced maximum Hessian eigenvalue and lower $\mathcal{L}_{\mathrm{LPF}}$. This provides empirical evidence that the benefits of DML scale naturally to tasks with 3 modalities, consistently outperforming both unimodal baselines and SML.

Table 10: **Performance Results on UR-Funny datasets**

| Metrics | Visual | Audio | Text | SML | DML |
|---------|--------|-------|------|-----|-----|
| Accuracy ($\uparrow$) | 59.42 | 59.28 | 54.15 | 62.57 | **63.78** |
| $\lambda_{\max}$ ($\downarrow$) | 1.3440 | 1.5302 | 1.4437 | 1.0934 | **1.0920** |
| $\mathcal{L}_{\mathrm{LPF}}$ ($\downarrow$) | 0.7226 | 0.6823 | 0.7186 | 0.6717 | **0.6705** |

## C.5 LOSS LANDSCAPE VISUALIZATION

In this section, we provide additional loss landscape visualizations following Li et al. (2018) for the datasets used in our experiments, Kinetics-Sounds, CREMA-D and UPMC-Food101, which corroborate our initial hypotheses.

For instance, the multimodal learning loss landscape (Figure 6) exhibits a flatter minima compared to the corresponding unimodal learning landscape (Figure 6a). Furthermore, DML (Figure 6c) demonstrates a smoother loss landscape relative to SML (Figure 6b), consistent with our expectations. Similar trends are observed across other datasets such in CREMA-D and UPMC-Food101: multimodal learning consistently produces flatter minima compared to unimodal learning. While the difference between DML and SML is marginal, DML still exhibits a slightly flatter landscape. To complement these qualitative observations with quantitative evidence, Table 4 and Figure 4 report generalization metrics of landscape flatness, which consistently align with our hypotheses.

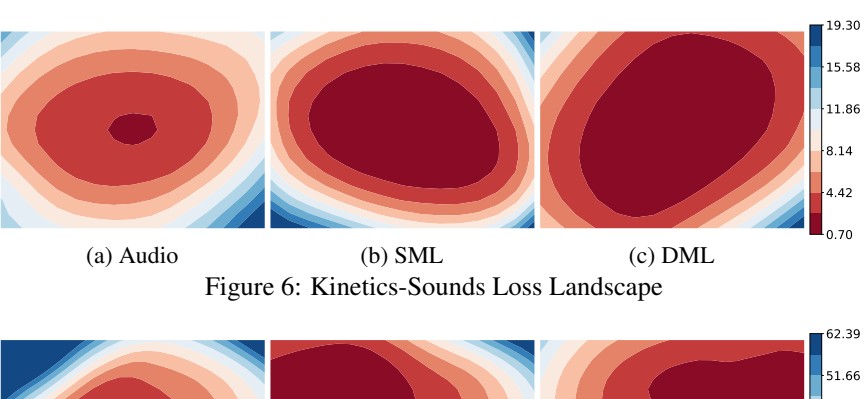

|     (a) Audio      |      (b) SML      |      (c) DML      |
|--------------------|-------------------|-------------------|

Figure 6: Kinetics-Sounds Loss Landscape

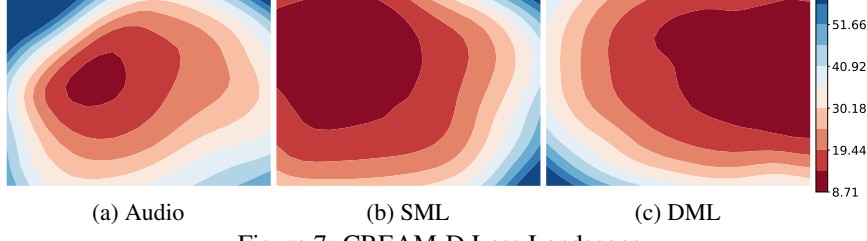

|     (a) Audio      |      (b) SML      |      (c) DML      |
|--------------------|-------------------|-------------------|

Figure 7: CREAM-D Loss Landscape

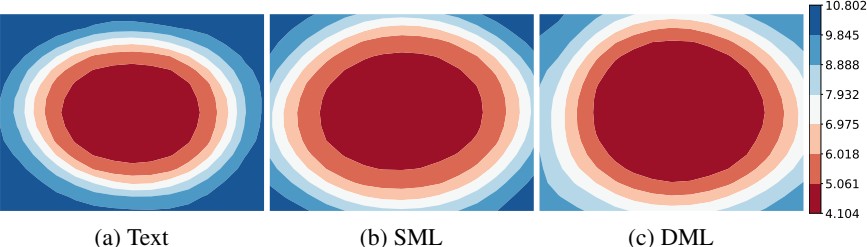

|     (a) Text       |      (b) SML      |      (c) DML      |
|--------------------|-------------------|-------------------|

Figure 8: UPMC-Food101 Loss Landscape

## C.6 Generalization Performance: Adversarial Robustness Evaluation

In this section, we evaluate the generalization performance of the models. Specifically, we present the results of adversarial perturbation experiments conducted on various multimodal datasets.

**Adversarial Robustness** Adversarial robustness evaluation provides an alternative lens on generalization performance by examining model behavior under worst-case perturbations, where each data point is transformed into its most challenging variant. In this paper, we employ the $\ell_\infty$-Norm Projected Gradient Descent (PGD) adversarial attack—a widely used method for evaluating adversarial robustness, formulated as follows:

$$\mathbf{x}_{t+1} = \Pi_{\mathcal{B}_\epsilon(\mathbf{x})}\left(\mathbf{x}_t + \alpha \cdot \text{sign}\left(\nabla_{\mathbf{x}}\mathcal{L}(f(\mathbf{x}_t), y)\right)\right)$$

where $\mathbf{x}_t$ denotes the perturbed input at iteration $t$ (with a total of $T$ iterations), $\mathcal{L}$ is the loss function, $f$ is the model, $y$ is the true label, $\alpha$ is the step size, and $\Pi_{\mathcal{B}_\epsilon(\mathbf{x})}(\cdot)$ is the projection operator onto the $\ell_\infty$-ball of radius $\epsilon$ centered at the original input $\mathbf{x}$. We adopt a modified version of the method from prior work (Yang et al., 2021), with detailed hyperparameter configurations provided in Table 11. For evaluation, we quantify robustness using the drop rate relative to clean accuracy:

$$\text{Drop Rate} = \frac{\text{Clean Accuracy } - \text{ Perturbed Accuracy}}{\text{Clean Accuracy}} \times 100$$

Table 11: **PGD Hyperparameter Settings**

|  | Image / Visual | Spectrogram$^\star$ (Audio) |
|---|---|---|
| Step $T$ | 5 | 10 |
| $\epsilon^\dagger$ | 0.1 | 0.005 |
| $\alpha^{\dagger\dagger}$ | 0.001 | 0.001 |

$^\dagger$: Perturbation Range  $^{\dagger\dagger}$: Adversarial Learning Rate  $^\star$: Spectrogram on KS, AVMNIST & CREMA-D

**Adversarial Evaluation Results** Tables 12 present the adversarial robustness results across all evaluated multimodal datasets under PGD attacks. Overall, unimodal models show pronounced vulnerability to adversarial perturbations. For instance, on AVMNIST (Table 12b), the vision-only model suffers a severe accuracy drop of 76.78%, while the audio-only model drops by 51.15%. Comparable patterns emerge across other datasets: for example, in CREMA-D (Table 12c), the audio-only model drops 33.58% accuracy under perturbation, while in Kinetics-Sounds (Table 12a) the image-only model experiences an 81.57% decrease. These consistent trends highlight the limited robustness of unimodal baselines compared to multimodal models. Moreover, across all datasets, DML consistently outperforms SML under adversarial perturbations, showing smaller accuracy drops for both modalities (e.g., [SML vs. DML] on Kinetics-Sounds, [12.64% vs. 14.34%] for audio and [34.38% vs. 35.19%] for image). This indicates that DML's more integrated representation learning yields enhanced robustness and better generalization compared to simpler fusion strategies.

For the UPMC-Food101 dataset, we have not applied adversarial perturbations to the text modality, as PGD attacks on text models are generally considered less natural and less interpretable. Instead, we restricted adversarial evaluations to the image modality. As reported in Table 12d, perturbations applied to the image modality reveal markedly greater robustness under multimodal learning compared with unimodal learning. Although the improvements of DML over SML are modest (only 0.27% gap), DML consistently exhibits smaller performance degradation and higher absolute accuracy. These findings suggest that multimodal learning not only converges to flatter minima but also achieves enhanced robustness and generalization relative to unimodal approaches.

Table 12: **Multimodal Adversarial Robustness Evaluation.** Performance is measured by the drop rate (↓) in accuracy compared to the Clean Accuracy after applying adversarial perturbations to individual modalities. A *lower drop rate* indicates *enhanced robustness or generalization performance.*

(a) **Kinetics-Sounds Adversarial Evaluation**

| **KS** | Clean Accuracy | Modality $i$ Perturbation | Modality $j$ Perturbation |
|---|---|---|---|
| Modality $i$: [A] | 51.95 | 39.73 (23.52% ↓) | – |
| Modality $j$: [I] | 45.70 | – | 8.42 (81.57% ↓) |
| SML | 62.95 | 53.92 (14.34% ↓) | 40.80 (35.19% ↓) |
| DML | 65.89 | **57.56 (12.64% ↓)** | **43.24 (34.38% ↓)** |

(b) **AVMNIST Adversarial Evaluation**

| **AVMNIST** | Clean Accuracy | Modality $i$ Perturbation | Modality $j$ Perturbation |
|---|---|---|---|
| Modality $i$: [I] | 64.98 | 15.09 (76.78% ↓) | – |
| Modality $j$: [A] | 41.84 | – | 20.44 (51.15% ↓) |
| SML | 70.30 | 25.81 (63.29% ↓) | 35.85 (49.00% ↓) |
| DML | 71.69 | **29.91 (58.28% ↓)** | **37.79 (34.68% ↓)** |

(c) **CREMA-D Adversarial Evaluation**

| **CREMA-D** | Clean Accuracy | Modality $i$ Perturbation | Modality $j$ Perturbation |
|---|---|---|---|
| Modality $i$: [A] | 57.30 | 38.06 (33.58% ↓) | – |
| Modality $j$: [V] | 49.16 | – | 36.56 (25.63% ↓) |
| SML | 59.62 | 41.30 (30.73% ↓) | 48.80 (18.15% ↓) |
| DML | 59.77 | **42.96 (28.12% ↓)** | **49.34 (17.45% ↓)** |

(d) **UPMC-Food101 Adversarial Evaluation**

| **Food101** | Clean Accuracy | Modality $i$ Perturbation | Modality $j$ Perturbation |
|---|---|---|---|
| Modality $i$: [T] | 86.38 | – | – |
| Modality $j$: [I] | 64.68 | – | 6.86 (89.39% ↓) |
| SML | 91.48 | – | 85.57 (6.46% ↓) |
| DML | 92.84 | – | **87.09 (6.19% ↓)** |

# D DISCUSSIONS

## D.1 EMPIRICAL VALIDATION OF THE ASSUMPTION 1

Our theoretical analysis focuses on late-fusion architectures, which are widely used in multimodal learning. We adopt the approximate translation-invariance (ATI) assumption (Assumption 1), which postulates that the fusion network exhibits near-invariant behavior under small shifts in one modality. Yet this is an assumption, it still warrants careful consideration and empirical validation. We analyze four widely used fusion mechanisms: Additive Fusion, Concatenation with linear MLP, Concatenation with non-linear MLP, and Cross-Attention (on infinite dimension) to determine the extent to which satisfies the ATI assumption:

- **Additive Fusion**: In this case, the assumption holds exactly with $\boldsymbol{\alpha} = \mathbf{1}$, as

$$\phi(\mathbf{u}, \mathbf{v} + \boldsymbol{\tau}) = \phi(\mathbf{u}, \mathbf{v}) + \boldsymbol{\tau}$$

- **Concatenation + Linear MLP**: Let $\phi(\mathbf{u}, \mathbf{v}) = W[\mathbf{u} \mid \mathbf{v}] + \mathbf{b}$, where $W = [W_{\mathbf{u}} \mid W_{\mathbf{v}}] \in \mathbb{R}^{d \times (d_i + d_j)}$. When a small perturbation $\boldsymbol{\tau}$ is applied to modality $\mathbf{u}$, the output becomes:

$$\phi(\mathbf{u}, \mathbf{v} + \boldsymbol{\tau}) = \phi(\mathbf{u}, \mathbf{v}) + W_{\mathbf{v}}\boldsymbol{\tau}$$

  Thus, the influence of $\boldsymbol{\tau}$ can be interpreted as a linearly predictable transformation. By defining $\boldsymbol{\alpha} := \mathrm{diag}(W_{\mathbf{v}})$, this linear effect can be approximated in the ATI form as:

$$\phi(\mathbf{u}, \mathbf{v} + \boldsymbol{\tau}) \approx \phi(\mathbf{u}, \mathbf{v}) + \boldsymbol{\alpha} \odot \boldsymbol{\tau}.$$

- **Concatenation + Non-linear MLP**: For deeper nonlinear networks (e.g., including ReLU activations), the ATI property holds approximately through a first-order Taylor expansion:

$$\phi(\mathbf{u}, \mathbf{v} + \boldsymbol{\tau}) \approx \phi(\mathbf{u}, \mathbf{v}) + J_\phi \cdot \boldsymbol{\tau} + R(\|\boldsymbol{\tau}\|^2)$$

  where $J_\phi := \frac{\partial \phi}{\partial \mathbf{v}} \in \mathbb{R}^{d \times d_j}$ is the Jacobian of $\phi$ with respect to $\mathbf{v}$. Moreover, $R(\|\boldsymbol{\tau}\|^2)$ is the higher-order remainder term satisfying $\mathcal{O}(\|\boldsymbol{\tau}\|^2)$ as $\|\boldsymbol{\tau}\|^2 \to 0$. This expansion shows that, under mild smoothness assumptions on $\phi$, the output shift is approximately linear in $\boldsymbol{\tau}$, and thus the ATI assumption: $\phi(\mathbf{u}, \mathbf{v} + \boldsymbol{\tau}) \approx \phi(\mathbf{u}, \mathbf{v}) + \boldsymbol{\alpha} \odot \boldsymbol{\tau}$, holds approximately if we let $\boldsymbol{\alpha} := \mathrm{diag}(J_\phi)$, i.e., the diagonal of the Jacobian. It provides an approximation to the shift effect that aligns with the elementwise form of the ATI assumption.

- **Cross-Attention**: Consider a single cross-attention block where $\mathbf{u}$ provides the queries and $\mathbf{v}$ provides the keys and values:

$$\phi(\mathbf{u}, \mathbf{v}) = \mathrm{Attn}(Q\mathbf{u}, K\mathbf{v}, V\mathbf{v}) = \mathrm{softmax}\left(\frac{(Q\mathbf{u})(K\mathbf{v})^\top}{\sqrt{d_k}}\right) V\mathbf{v}.$$

  A perturbation $\boldsymbol{\tau}$ to $\mathbf{v}$ induces changes $K(\mathbf{v} + \boldsymbol{\tau}) = K\mathbf{v} + K\boldsymbol{\tau}$ and $V(\mathbf{v} + \boldsymbol{\tau}) = V\mathbf{v} + V\boldsymbol{\tau}$, affecting both weights and values. Because the attention operator is smooth, we can write the first-order Taylor expansion:

$$\phi(\mathbf{u}, \mathbf{v} + \boldsymbol{\tau}) = \phi(\mathbf{u}, \mathbf{v}) + J_\phi \boldsymbol{\tau} + R(\|\boldsymbol{\tau}\|^2),$$

  where

$$J_\phi := \frac{\partial \phi}{\partial (K\mathbf{v})} K + \frac{\partial \phi}{\partial (V\mathbf{v})} V \in \mathbb{R}^{d \times d_j}$$

  is the Jacobian w.r.t. $\mathbf{v}$. The *over-parameterized (infinite-width) lazy training paradigm* (Jacot et al., 2018; Yang, 2020; Hron et al., 2020) provides the justification for dropping the high order $R$ term, where $\mathcal{O}(\|\boldsymbol{\tau}\|^2)$ as $\|\boldsymbol{\tau}\|^2 \to 0$. These theories confirm that even complex architectures like Transformers or attentions are dominated in case of infinite dimension space by their first-order approximation, as $J_\phi$ remains stable. Therefore, this allows to approximate as:

$$\phi(\mathbf{u}, \mathbf{v} + \boldsymbol{\tau}) \approx \phi(\mathbf{u}, \mathbf{v}) + J_\phi \boldsymbol{\tau}.$$

  Thus, the ATI form holds approximately by defining $\boldsymbol{\alpha} := \mathrm{diag}(J_\phi)$.

These derivations indicate that our ATI assumption is consistent with the behavior of recent complex models under the standard theoretical framework of modern deep learning.

Next, to empirically validate the ATI assumption, we introduced a small perturbation $\tau$ to $\mathbf{v}$ in each input pair $(\mathbf{u}, \mathbf{v}) \in \mathbb{R}^d$. For the experiments, we employed a shallow network for the linear MLP, a two-layer network with a ReLU activation for the non-linear MLP, and a 8-head multi-head attention block for the cross-attention. We define the feature norm as $\|\phi(\mathbf{u}, \mathbf{v})\|_2$, invariance error as $\Delta_{\texttt{error}} = \|\phi(\mathbf{u}, \mathbf{v} + \tau) - \phi(\mathbf{u}, \mathbf{v}) - \alpha \odot \tau\|_2$ and the relative error to quantify its effect on the feature magnitude as $\epsilon_{\texttt{rel}} = \|\Delta_{\texttt{error}}\|_2 / \|\phi(\mathbf{u}, \mathbf{v})\|_2$.

Table 13: **Testing of ATI Assumption for Various Fusion Networks**

| **Metric** | **Additive** | **Linear MLP** | **Non-Linear MLP** | **Cross-Attention** |
|---|---|---|---|---|
| Feature Norm | 1.4103 | 0.6220 | 0.6839 | 0.4267 |
| Invariance Error | 0 | 0.0047 | 0.0067 | 0.0051 |
| $\epsilon_{\texttt{rel}} \times 100$ | 0 | 0.76 % | 0.98 % | 1.20 % |

The results in Table 13 demonstrate that all fusion strategies exhibit only minor deviations from the linear approximation, with $\epsilon_{\texttt{rel}}$ quantifying the relative magnitude of these deviations. As expected, additive fusion satisfies ATI exactly. Both MLP-based variants show small errors, indicating that the ATI assumption holds approximately, even in non-linear network. Notably, the cross-attention fusion also exhibits only a minor relative error, despite its more complex nonlinear interactions. Taken together, these findings provide empirical support for the validity of our ATI-based theoretical framework across a wide range of practical architectures, and the approximation is expected to become increasingly accurate in the infinite-dimensional regime.

### D.2 ON THE VALIDITY OF THE ASSUMPTION 2

In this section, we address the validity of Assumption 2, which posits conditional independence between modalities. At first glance, this assumption may appear restrictive or unrealistic in real-world multimodal settings. Furthermore, it raises the question of whether the underlying distributions for SML and DML are theoretically identical under this assumption. However, as we clarify below, this formulation is neither impractical nor incompatible with standard learning pipelines. Rather, it naturally aligns with the generative objective of multimodal learning and provides a coherent framework for distinguishing the sampling behavior of DML from that of SML.

**The Role of $y$ in Multimodal Learning**    The goal of multimodal learning is not to model incidental correlations between modalities, but to learn a mapping from input space to the underlying hypothesis space. Under this perspective, Assumption 2 characterizes a general and reasonable setting: once the true hypothesis $y$ is specified, individual modalities are not required to be highly correlated with each other; rather, they serve as independent evidence supporting that hypothesis. This viewpoint is consistent with a wide range of multimodal tasks in which modalities serve as complementary but conditionally independent with given hypothesis space $y$, whether $y$ corresponds to a label space or a semantic space. Empirically, our experiments across diverse tasks, such as multimodal classification (Table 1) and image–text retrieval (Table 7), demonstrate that DML setting is more effective.

**Sampling Space: SML vs. DML**    From this standpoint, the key distinction between SML and DML lies in their sampling spaces, not in the distributions they target. Under Assumption 2, DML samples from the full conditional product $p(x_i, x_j \mid y) = p(x_i \mid y)p(x_j \mid y)$ thereby covering *all valid cross-modal pairings* consistent with hypothesis $y$. In contrast, SML samples only from the fixed paired set—the finite set of modality pairs that co-occur in the dataset. This excludes all cross-pairings that do not appear in the raw data, even though such combinations are valid $p(x_i, x_j \mid y)$. Consequently, both methods sample from the same underlying distribution under Assumption 2. The difference is that SML explores only a narrow subset of the admissible sampling space, whereas DML explores it fully. In other words, SML corresponds to a special case of DML in which sampling is restricted to a fixed paired subset of the conditional product space. Therefore, while there is no difference in the target distribution, DML explores more comprehensively than SML.

In summary, we argue that the conditional independence assumption is not only plausible in practical multimodal settings, but in fact yields a more principled and generalizable learning framework.

