# OpenReview forum: "Understanding Multimodal Learning: A Loss Landscape Smoothness Perspective"
_ICLR.cc/2026/Conference — Submitted to ICLR 2026_

### Official Review · Reviewer_Hoxa · 2025-10-29

**Soundness:** 3
**Presentation:** 3
**Contribution:** 3
**Rating:** 8
**Confidence:** 3

**Summary:**

The authors propose that multimodal learning leads to better performance than unimodal learning due to the presence of label-dependent variance that smooths the loss landscape and leads to flatter minima. They argue this theoretically by proving that the multimodal loss can be expressed as a convolution of the unimodal loss with a "scaled-shift kernel" derived from the second modality, which guarantees a smoother loss landscape. This motivates a very simple training procedure which shuffles the pairs of inputs which belong to the same level. This procedure leads to consistent and sometimes significant improvements on multiple multimodal benchmark datasets.

**Strengths:**

* Distributed Multimodal Learning (DML) is extremely simple to implement, introduces no hyperparameters, and performs well empirically across a diverse set of multimodal datasets.
* The choice of datasets spans three pairs of modalities: audio/vision, image/audio, image/text. This makes the conclusions much more general and believable compared to e.g. just considering multiple image/text datasets.
* DML consists of shuffling pairs of inputs which correspond to the same label. This introduces a label-conditioned variance, which is consistent with the authors' theory.

**Weaknesses:**

* The experimental results can be made more robust by comparing across at least two architectures for each dataset.
* The datasets and network architectures are relatively small-scale, and it would be interesting to see whether DML improves results on e.g. larger image/text datasets with larger vision and text encoders.

**Questions:**

Do you think the assumption of conditional independence between the inputs given the target is reasonable? Your empirical results seem to suggest so, but I would be interested to see this discussed in the paper. There is previous work in the literature that suggests the opposite conclusion, which is that the inter-modal dependency is precisely why multimodal learning is advantageous over unimodal learning [1].

[1] Jointly Modeling Inter- & Intra-Modality Dependencies for Multi-modal Learning, Madaan et al, NeurIPS 2024.

---

> ### Author Response · Authors · 2025-11-20
> **Response to Reviewer Hoxa**
>
> ### **Dear Reviewer Hoxa**
>
> We thank the reviewer for pinpointing this specific ambiguity. Taking this feedback into account, we have substantially revised our theoretical discussion to ensure clarity and fully address the primary concerns:
>
> &nbsp;
>
> >### **Weakness 1. & Weakness 2: More Architectures and Scalability of Datasets & Models**
>
> We thank the reviewer for raising the important point regarding architectural diversity and the scalability of datasets and models. We fully agree that verifying DML across multiple architectures and larger-scale settings is valuable for demonstrating robustness.
>
> Yet in practice, prior works in multimodal learning typically adopt a *single* backbone architecture (most commonly ResNet-based architecture) when evaluating multimodal objectives, primarily because the goal is to isolate the effect of the learning objective rather than the backbone capacity. Therefore, our main experiments are conducted on ResNet-based encoders.
>
> To further address the reviewer’s concern, we would like to highlight that our Appendix C.1. already includes **CLIP-based training experiments**, which utilize significantly larger-scale vision and text encoders and datasets. These results demonstrate that DML continues to provide consistent and often improved performance even when applied on top of large pretrained encoder pairs. We believe this provides strong evidence that the proposed method scales to more powerful architectures beyond small or mid-sized models. Here, we show the results from the Appendix C.1 :
>
> | Method | I2T R@1 | I2T R@5 | I2T R@10 | T2I R@1 | T2I R@5 | T2I R@10 |
> | :--- | :---: | :---: | :---: | :---: | :---: | :---: |
> | **SML (Fixed Pair)** | 87.34 | 97.10 | **99.56** | 74.19 | 89.77 | 93.53 |
> | **DML (Shuffled Pair)** | **89.10** | **97.56** | 99.31 | **76.10** | **90.51** | **93.89** |
>
> We hope these clarifications address the reviewer’s concerns. Overall, while our main experiments follow the standard, widely adopted ResNet-based evaluation protocol, our additional CLIP experiments demonstrate that DML generalizes well to large-scale architectures and datasets.
>
> &nbsp;
>
> >### **Question 2. Validity and Justification of Assumption 2**
>
> We thank the reviewer for this insightful criticism. In our understanding, the reviewer points to prior work suggesting that inter- or intra-modality dependencies can be crucial in multimodal learning, especially for tasks such as VQA, where cross-modal interactions directly affect predictive performance. We fully agree with this perspective. However, we would like to clarify the role of **Assumption 2 (Conditional Independence)**. This assumption should not be interpreted as a universal claim that multimodal data are literally conditionally independent in all scenarios. Instead, it serves as a **theoretical approach** to propose our main point of view: when the modalities share a hypothesis space $y$, it becomes possible to construct a training strategy that samples from $p(x_i \mid y)$ and $p(x_j \mid y)$ independently, rather than being constrained to fixed pairs.
>
> This means that Assumption 2 does not contradict the main conclusions of prior work (e.g., I2M2 [1]), but instead represents a broader conceptual view that can cover a wide range of multimodal learning settings. In this sense, it is closer to a general formulation of multimodal learning. Additionally, we note that the SML approach (sampling each modality dependently) is a special case within the broader DML.
>
> To further support this point, we have already provided additional empirical results in Appendix C.1, where we evaluate DML in a contrastive learning setting. Even in datasets without explicit labels defining the hypothesis space, once we approximate this space through pseudo-labeling (UMAP +K-means), DML with stochastic pairing still achieves competitive or even better performance compared to methods relying strictly on fixed paired data, such as SML. Results on Appendix C.1 justify that our assumption also captures a practically useful regime in multimodal learning.
>
> For these reasons, we believe that Assumption 2 is both reasonable and applicable in practical settings such as multimodal classification and image–text retrieval. Moreover, it does not undermine prior work [1]; rather, by emphasizing the role of the shared hypothesis space $y$, it conceptually supports and generalizes main arguments of prior work.
>
> [1] Jointly Modeling Inter- & Intra-Modality Dependencies for Multi-modal Learning, Madaan et al, NeurIPS 2024.
>
> &nbsp;
>
> ==========================================================
>
> **We again thank the reviewer for their insightful comments. We have revised the manuscript to address their concerns and provide the necessary clarifications. We hope overall concerns from reviewer have been clarified.**

---

### Official Review · Reviewer_EYm2 · 2025-10-30

**Soundness:** 2
**Presentation:** 3
**Contribution:** 2
**Rating:** 4
**Confidence:** 4

**Summary:**

This paper examines multimodal learning from the perspective of landscape smoothness in loss functions. First, it points out that while multimodal learning has significant empirical advantages, its theoretical foundation is weak (existing theories rely on strong assumptions and lack empirical verification). Then, it proposes a theoretical framework based on convolutional smoothing. By defining notations for modal input, encoder, and fusion function, and combining the two assumptions of "approximate translation invariance of the fusion function" and "conditional independence of modalities under given labels," it derives the theorem that multimodal expected loss is the convolution of single-modal loss and scale-shifted kernel (Theorem 1), and the theorem that multimodal loss results in a smoother landscape (Theorem 2, reflected in the upper bound of the Hessian spectral norm and the effect of low-pass filtering in the frequency domain). Based on this theory, the paper further proposes a Distributional Multimodal Learning (DML) strategy, replacing traditional fixed-point pairing (SML) with random modal pairing within the same label space to enhance the convolutional smoothing effect. Finally, the results are validated on four multimodal datasets, including Kinetics-Sounds and AVMNIST. The results show that DML has superior performance (e.g., Kinetics-Sounds accuracy). The DML method outperforms SML and unimodal methods in several aspects,

**Strengths:**

1. The motivation is clear and convincing.

2. This paper fills the theoretical gap in multimodal learning; a rigorous "convolutional smoothing" framework is constructed.

3. The proposed DML is simple and easy to follow.

**Weaknesses:**

1. The assumption 2 of conditional independence seems impossible in multimodal learning. The multimodal data capture from the same object is naturally dependent. Especially, the multimodal image data, such as RGB and IR, when the position of the object in the RGB changes, it will also change in the IR image. Similarly, in video-audio data, where the visual action and sound are highly synchronized, a strong dependency still exists even after a label is given.

2. Besides, the ATI assumption holds only in shallow fusion architectures such as additive fusion and simple splicing + MLP, and its "approximate translation invariance" is difficult to satisfy for current mainstream architectures with stronger nonlinearity such as attention fusion and cross-modal Transformer.

3. The performance gains of DML are counterintuitive. If intra-class random perturbation can bring performance gains, then there is no need to collect matching multimodal data. Just collect arbitrary unimodal data, classify it, and then randomly match within the same class.

4. How to extend DML to the task with three modalities?

**Questions:**

See the weakness

---

> ### Author Response · Authors · 2025-11-20
> **Response to Reviewer EYm2 (Part 1/2)**
>
> ### **Dear Reviewer EYm2**
>
> We are grateful for the reviewer’s insightful assessment, which accurately identifies a lack of clarity in our theoretical discussion. In response, we have rigorously refined our analysis to resolve the points raised:
>
> &nbsp;
>
> >### **Weakness 1. On the Validity of the Assumption 2 (Conditional independence)**
>
> We first thanks to the reviewer providing insightful claims to our approaches.
>
> In our understanding, the reviewer argues that Assumption 2 (Conditional Independence) seems "impossible" in reality, citing the strong physical synchronization and dependency in data like RGB/IR or Video-Audio. In this paper, we view this in a different perspective; fundamental goal of deep learning objective $y$.
>
> It is true that modalities such as RGB–IR or video–audio exhibit strong physical correlations in the raw world. However, the goal of (multimodal) deep learning is not to model these incidental correlations, but to learn the mapping. Under this perspective, Assumption 2 ($p(x_i, x_j \mid y) = p(x_i \mid y)p(x_j \mid y)$) shows this as general setting of multimodal learning. When the true hypothesis $y$ is specified, each modality does not required to be highly correlated, but independent evidence about that hypothesis.
>
> This idea is clearer in settings like image–text retrieval, where many diverse captions correspond to the same underlying semantic label. In datasets like COCO or Flickr-30k, a single image $x_i$ (given $y$=dog) is paired with multiple, diverse captions ($x_j$). These captions are not physically dependent on each other in the same way RGB/IR pixels are; their only common link is the **hypothesis $y$** (the image content). This demonstrates a setting where the $x \rightarrow y$ goal is primary. Our DML aims to focus on mapping input data to hypothesis space.
>
> Therefore, we want to argue that Assumption 2 is not "impossible"; it is the theoretical foundation that both SML and DML upon. The crucial difference is how they sample from the resulting sampling space, based on hypothesis space. The "dependency" the reviewer observes is an artifact of SML's biased, subset sampling of overall sampling space, which encourages the model to learn spurious $x_i \leftrightarrow x_j$ shortcuts. DML, in contrast, also satisfies this assumption but does so by sampling from the entire, high-variance distribution $p(x_i \mid y)p(x_j \mid y)$. In other words, **SML is just one of case that could be occured on DML.**
>
> **We revised on Appendix D.2, for the justification of Assumption 2.**
>
> &nbsp;
>
> >### **Weakness 2. Justification of the Assumption 1 (ATI)**
>
> We thank the reviewer for their insightful comment, which points to a key ambiguity in our current theoretical discussion.
>
> The reviewer correctly identifies that our current Appendix C.2 analyzes the linear approximation primarily from a **finite-dimensional perspective**. We concede that this view alone, as currently written, may appear insufficient to fully justify the assumption for strongly non-linear mechanisms like cross-attention.
>
> Yet, the theoretical foundation can be naturally extended by considering the **infinite-width limit of modern over-parameterized architectures [1,2,3]**, a standard paradigm for analyzing contemporary deep networks. In this "lazy training" (NTK) regime, it is a well-established theoretical result that even complex architectures like **Transformers including attention mechanism can be linearly approximated** (i.e., they are dominated by their first-order Taylor expansion). Prior work has shown this explicitly for attention layers [1, 2], and our current finite-dimensional analysis is consistent with this broader framework.
>
> **Testing of Assumption 1 for cross-attention fusion networks**
>
> | Feature Norm (Scale) | Average Invariance Error | Error Percentage |
> | --- | --- | --- |
> | 0.4267 | 0.0051 | 1.1952 |
>
> Thus, Assumption 1 is still a valid approximation backed by this extensible theoretical framework. To further validate this theoretical argument, we have added new experiments using a **cross-attention fusion network.** As the result, it also show a minimal **approximation error**, empirically confirming our assumption's validity.
>
> We still acknowledge that our submission did not make this theoretical connection sufficiently clear, where we only view in finite dimension space. For the clarity, **we moved Appendix C.2 to Appendix D.1, as discussion part, to explicitly include this crucial discussion on the infinite-dimensional justification and added Cross-Attention part**, clarifying how our assumption 1 can be extends to complex networks.
>
> **[1]** Yang, G. "Tensor Programs II: Neural Tangent Kernel for Any Architecture." ICLR 2021.\
> **[2]** Hron, J., et al. "Infinite Attention: NNGP and NTK for Deep Attention Networks." ICML 2020.\
> **[3]** Jacot, A., et al. "Neural Tangent Kernel." NeurIPS 2018.\

---

> ### Author Response · Authors · 2025-11-20
> **Response to Reviewer EYm2 (Part 2/2)**
>
> >### **Weakness 3. Intra-class random perturbation**
>
> First, we thank the reviewer for this excellent and thought-provoking question. We believe this point can be clarified most effectively by examining the problem from the perspective of the **hypothesis space**.
>
> We understand the reviewer’s suggestion as follows: if intra-class random perturbation alone leads to performance gains, then one might not need genuine multimodal datasets. Instead, one could collect arbitrary unimodal data, assign class labels ($y$), and randomly match samples within each class during training. However, our method—DML—is not intended to address the data-collection problem. Rather, as we argue under Weakness 1 and formalize in Assumption 2, DML is designed for the standard scenario of multimodal learning, where fixed paired data exist but may be limited or biased.
>
> The performance gains of DML do not arise from arbitrary random matching. This arises because DML expands the effective hypothesis space available during training by covering the overall sampling distribution. In other words, “same-class matching” is merely one simple instantiation of a broader principle: DML covers a larger sampling space than the SML approach, which is restricted to an underspecified subset of the joint distribution $p(x_i \mid y)p(x_j \mid y)$.
>
> Therefore, we would like to claim that the better performance of DML over SML is not counter-intuitive. It is, in fact, a highly intuitive consequence of employing a better, debiased training objective that captures more of the underlying data distribution.
>
> &nbsp;
>
> >### **Weakness 4. Extending DML to settings with 3 modalities datasets**
>
> Extending DML to three modalities is straightforward. The core idea naturally generalizes by assuming that the conditional independence structure extends across all modalities, i.e., $p(x_i \mid y)p(x_j \mid y)p(x_k \mid y) = p(x_i, x_j, x_k \mid y)$. This corresponds to the multimodal setting where each modality shares the same class-conditional semantics. Empirically, we validate this extension on the tri-modal UR-Funny dataset (text, audio, and vision).
>
> To empirically validate this extension, we evaluate DML on the **UR-Funny** dataset (text, audio, and vision), a benchmark containing **three** modalities. The results are summarized below:
>
> | UR-Funny | Modality i | Modality k | Modality k | SML | DML |
> | --- | --- | --- | --- | --- | --- |
> | Accuracy | 59.42 | 59.28 | 54.15 | 62.57 | **63.78** |
> | $\lambda_\max$ | 1.3440 | 1.5302 | 1.4437 | 1.0934 | **1.092** |
> | $\mathcal{L}_{\texttt{LPF}}$ | 0.7226 | 0.6823 | 0.7186 | 0.6717 | **0.6705** |
>
> These results demonstrate that DML continues to outperform SML even in the three modalities setting. Not only does DML improve accuracy by a clear margin (+1.21%), but it also achieves flatter minima, as reflected in both the reduced maximum Hessian eigenvalue and lower $\mathcal{L}_{\texttt{LPF}}$. This provides strong empirical evidence that the benefits of DML scale naturally to the 3-modality datasets, outperforming both unimodal models and even SML.
>
> **We have added this content to Appendix C.4 on the manuscript.**
>
> &nbsp;
>
> ==========================================================
>
> **We deeply appreciate the reviewer's thoughtful examination and constructive recommendations. We hope our responses have successfully addressed the reviewer's concerns and questions.**

---

> > ### Comment · Reviewer_EYm2 · 2025-11-26
> >
> > The reviewers acknowledged that in cross-modal retrieval tasks, the same label (y=dog) can have multiple associated images, but this appears to be unimodal (different x_i from X_i) rather than multimodal.
> >
> > The paper describes p(x_i,x_j|y)=p(x_i|y)p(x_j|y), where x_i and x_j come from different modalities but share the same label. This differs from the setup of cross-modal retrieval, so the explanation provided is not convincing.

---

> > > ### Author Response · Authors · 2025-11-26
> > >
> > > We sincerely thank the reviewer for the constructive feedback. However, we would like to clarify that the reviewer’s concern reflects a confusion of our assumption, which is intended to model a multimodal learning rather than a unimodal setting. The reviewer claim that
> > >
> > > >“the same label ($y$=dog) can have multiple associated images, which appears to be unimodal (different $x_i$ from $X_i$ ) rather than multimodal.”
> > >
> > > - First, we apologize for any confusion during the rebuttal. As mentioned, the example “$y$=dog” on cross-modal retrieval datasets was intended solely for illustration. The purpose was to explain the **pairing mechanism in multimodal learning**, emphasizing that multiple valid pairings between modalities exist, **rather than implying a strict 1-to-1 correspondence**.
> > > - To clarify, our example involved one image paired with five textual captions. The main point is that $y$ is not restricted to a label space, but also could be the semantic content of the dataset. For instance, given an image of a dog playing frisbee, valid captions might include “The dog is playing frisbee” or “There is a dog on the field.” In this sense, **the hypothesis space extends beyond discrete labels and captures the semantics of the modalities**.
> > > - Our approach highlights that many recent multimodal learning benchmarks rely on fixed, paired data. In contrast, DML **leverages sampling within the hypothesis space**, which improves performance, robustness, and flatness of the optimization landscape.
> > > - Therefore, given Assumption 2, $p(x_i, x_j | y) = p(x_i | y) p (x_j | y)$, should be interpreted as a general form of multimodal learning, **$x_i$ and $x_j$ is from given datasets based on task-specific hypothesis space $y$**, rather than a restriction to unimodal. Moreover, this assumption have proved via DML that it is valid and useful for modeling the relationships between heterogeneous modalities conditioned on a shared semantic.
> > >
> > > To further clarify the reviewer’s concern and support above responses, we have **provided experimental results in Appendix C.1 (lines 1008–1043)**, where it has been already on the first submission. Specifically, we evaluated our method on **cross-modal retrieval tasks** and observed consistent performance improvements from +0.46% to +1.91%. These empirical results support that the hypothesis space $y$ is not required to be restricted to a discrete label space, but any possible hypothesis space. Consequently, we posit that **our assumption remains sufficiently valid for capturing underlying semantic factors, label information, and other task-specific attributes in practical multimodal learning settings**.
> > >
> > > &nbsp;
> > >
> > > We appreciate again the reviewer for their careful evaluation and questions. We hope that our responses have adequately addressed the questions and concerns raised.

---

### Official Review · Reviewer_LpsV · 2025-10-30

**Soundness:** 2
**Presentation:** 2
**Contribution:** 3
**Rating:** 4
**Confidence:** 4

**Summary:**

This paper examines multimodal learning from the perspective of **loss landscape smoothness**. The authors argue that combining modalities implicitly produces a *convolutional smoothing* effect on the loss surface, leading to flatter minima and improved generalization. They formalize this intuition through two theorems showing that multimodal losses can be expressed as convolutions of unimodal losses and that the resulting Hessian has a smaller spectral norm. Building on this idea, they propose **Distributional Multimodal Learning (DML)**, which randomly pairs modalities within the same class to further enhance smoothness. Experiments on several small- to medium-scale multimodal datasets (Kinetics-Sounds, AVMNIST, CREMA-D, and UPMC-Food101) show modest improvements in both accuracy and flatness.

**Strengths:**

1. Section 3.2 presents interesting and well-grounded theoretical results showing that multimodal losses tend to be smoother than unimodal ones. The argument is supported by clear intuition in the frequency domain and effectively summarized in Figure 2.
2. The experimental setup includes multiple datasets and evaluation metrics, demonstrating the benefits of multimodal training over unimodal baselines and, additionally, of DML over standard multimodal learning (SML).
3. The main practical contribution — the DML framework — appears to be novel despite its conceptual simplicity. It provides a straightforward and elegant form of data augmentation that is easy to implement and intuitively well motivated.

**Weaknesses:**

## Major
1. **Weak connection between theory and method.**
   Sections 3.2 and 3.3 feel somewhat disconnected. If DML is designed to improve the smoothness of the loss, this connection should be made more explicit. At present, it is unclear *why* random pairing within a class should theoretically enhance smoothness. Providing further intuition or a formal link between the proposed theory and the DML formulation would considerably strengthen the paper.

2. **Limited baseline comparison.**
   While the experimental setup covers several datasets and metrics, the baseline selection is relatively narrow. It would be important to compare against well-established methods known to improve generalization or flatness, such as:
   - **Stochastic Weight Averaging (SWA)** [1]
   - **Sharpness-Aware Minimization (SAM)** [2]
   - Regularization approaches like **Dropout** [3] or **Label Smoothing** [4]
   Including these would provide a stronger empirical foundation for the claims about smoothing and generalization.

I find the theoretical perspective and the overall idea of the paper both interesting and promising. However, the work needs to improve in terms of formal rigor—particularly by addressing the two key methodological issues mentioned above: (i) establishing a clearer connection between the theoretical analysis and the proposed DML method, and (ii) providing stronger comparisons against well-known baselines. These aspects are pivotal for ensuring the coherence and credibility of the contribution. If these limitations are addressed, I would be willing to increase my rating.

---

## Minor
1. *Assumption 1* is effectively a **definition**, though it is presented as an assumption. The wording should reflect this distinction more clearly.
2. In Table 1, the title should read **“Classification accuracy on multimodal datasets”** rather than “Classification results on multimodal datasets.”

**Questions:**

1. Would combining DML with some of the baseline methods mentioned above (e.g., SAM, SWA, Dropout, or Label Smoothing) further improve performance?
2. Do you believe that DML’s improvements primarily stem from enhanced smoothing, or could they be explained simply as a stronger form of data augmentation?

---

> ### Author Response · Authors · 2025-11-20
> **Response to Reviewer LpsV (Part 1/2)**
>
> ### **Dear Reviewer LpsV**
>
> We sincerely thank the reviewer for their constructive feedback. We appreciate the opportunity to clarify the ambiguity in our theoretical discussion and have addressed the core concerns as follows:
>
> &nbsp;
>
> >### **Weakness 1. Weak connection between theory and method.**
>
> The reviewer points out that the connection between our theory (smoothness) and our method (DML) appears weak. We agree that this link is a critical part of our paper, and we appreciate the opportunity to clarify it. The reviewer’s core question is: “Why should random pairing (DML) theoretically enhance smoothness?” Our DML method is, in fact, the practical realization of the smoothing mechanism described in our theory, and we will make this connection explicit, as it directly relates to Theorem 2 and Remarks 1 and 2.
>
> As stated in our Theorem 2 and elaborated in Remark 2 (Frequency Domain), the smoothing mechanism of multimodal learning is mathematically equivalent to a convolution that acts as a low-pass filter ($\widehat{\mathcal{L}}\_{\text{multi}} = \widehat{\mathcal{L}}\_{\text{uni}} \cdot \widehat{\mathcal{K}}$). Remark 2 explicitly states that this filter's effectiveness, its ability to suppress high-frequency components, depends on the kernel $\mathcal{K}$ having "sufficient variance". In our framework, this critical kernel $\mathcal{K}$ is the distribution of the "partner" modality, $p(x_j|y)$.
>
> In a standard SML (Fixed Pair) setting, however, this kernel is not a broad distribution. For any given $x_i$, its partner $x_j$ is deterministically fixed, creating a "kernel" with (near-)zero variance that acts as a dirac delta function. Based on Remark 2, such a zero-variance kernel provides minimal to no smoothing effect and fails to suppress the high-frequency "sharpness" of the unimodal loss.
>
> In contrast, our DML (Random Pairing) is explicitly designed to solve this by maximizing the kernel variance. By sampling $x_j$ from the entire marginal distribution $p(x_j|y)$, DML ensures that the kernel $\mathcal{K}$ has the largest possible variance (and conceptually, the highest entropy) that the data can provide. This high-variance kernel is precisely what Remark 2 requires to act as a powerful low-pass filter, effectively smoothing the loss landscape by suppressing high-frequency components.
>
> Therefore, the connection between our theory and method is, in fact, the central novelty of our work. The reviewer's "weak connection" is the key insight:
>
> - **Theorem 2:** High **variance** $\rightarrow$ improves **smoothness**.
> - **DML:** maximize the **variance**.
> - **SML:** provides the minimal **variance**.
>
> This observation clarifies why DML successfully engages the smoothing derived by theorem 2, whereas better than SML.
>
> &nbsp;
>
> >### **W2. & Q1. Limited baseline comparison; Smoothness approach**
>
> Following the reviewer's recommendation, we conducted additional experiments to benchmark DML against Label Smoothing (LS), Stochastic Weight Averaging (SWA), and Sharpness-Aware Minimization (SAM), all applied to our SML, where all results are averaged over 3 times. These conjunctions with SML can be understood as the baselines of multimodal training of LS, SWA, and SAM.
>
> As shown in the following results, DML almost shows better or equal performance, except CREMA-D. This means that DML is a simple but powerful empirical suggestion.
>
> |  | KS | AVMNIST | CREMA-D | UPMC Food101 |
> | --- | --- | --- | --- | --- |
> | SML | 62.95 | 70.3 | 59.62 | 91.48 |
> | SML + LS | 63.03 | 70.44 | 59.53 | 91.79 |
> | SML + SWA | 64.88 | 69.24 | 59.01 | 92.08 |
> | SML + SAM | 64.19 | 70.91 | **60.51** | 91.76 |
> | DML | **65.89** | **71.69** | 59.77 | **92.84** |
>
> Across most datasets, DML achieves equal or superior performance relative to these techniques, with CREMA-D being the only exception. These results indicate that DML provides a simple yet consistently strong empirical benefit compared to standard smoothing. **We added this extended comparison at Appendix C.3.**

---

> ### Author Response · Authors · 2025-11-20
> **Response to Reviewer LpsV (Part 2/2)**
>
> >### **Question 2: Data Augmentation (DA) vs. DML**
>
> Yes, we believe that the performance gains of DML primarily arise from its enhanced smoothing effect, as supported by both the qualitative and quantitative analyses presented in the main paper, rather than from data augmentation (DA).
>
> From another perspective, although DML may be viewed as a form of DA in a broad concept, prior studies have shown that the benefits of DA generally result from driving optimization toward flatter minima, which leads to tighter generalization bounds [1]. This aligns with our interpretation that DML is effective because it induces a stronger smoothing mechanism, not because it functions as a conventional augmentation method.
>
> **To clarify this, we have provided a detailed discussion in Appendix C.2 of the revised manuscript.**
>
> [1] Yoo, Weebum, and Sung Whan Yoon. "A Flat Minima Perspective on Understanding Augmentations and Model Robustness." Preprint (2025)
>
> &nbsp;
>
> >### **Minor Weaknesses**
>
> We appreciate the reviewer’s comments. Regarding Assumption 1, we intentionally present it as an assumption rather than a definition because the Approximately Translation-Invariant (ATI) property is not meant to define a general class of fusion networks, but to impose a structural condition required for our analysis. For this reason, we believe it is more appropriate to retain it as an Assumption rather than a Definition.
>
> We also thank the reviewer for pointing out the issue in Table 1. We have updated the table title to “Classification accuracy on multimodal datasets” accordingly.
>
> &nbsp;
>
> ==========================================================
>
> **We appreciate the reviewer's perceptive comments. We hope our responses have addressed reviewer's concerns and provided the necessary clarifications.**

---

> ### Comment · Reviewer_LpsV · 2025-11-25
>
> > Weakness 1: Clarification on DML and Variance
>
> I appreciate the clarification provided by the sentence: “By sampling from the entire marginal distribution, DML ensures that the kernel has the largest possible variance (and conceptually, the highest entropy) that the data can provide.” This effectively clears up my confusion, and I strongly recommend including this explanation in the revised manuscript. However, I suggest clarifying that this refers to the highest variance in the absence of data augmentations.
>
> To empirically validate the two main claims of the paper (and specifically Theorem 2), I propose the following experiment, which extends the analysis currently found in Appendix C.2. Given its importance, I believe this experiment should be included in the main text, potentially completing Table 2 and Figure 3:
>
> - Theorem 2 Validation: Since Theorem 2 posits that higher variance leads to higher smoothness, increasing the number of applied augmentations should theoretically increase smoothness. While Table 6 currently shows the effect of single augmentations, it would be compelling to show a cumulative trend: demonstrating that 3 augmentations result in better performance/smoothness than 2, and 2 better than 1, etc.
> - DML as Augmentation: As noted in my original review, I view DML as a form of augmentation. Since DML obtains the highest variance without explicit augmentation, I recommend including it in this comparison sequence to see how it complements the strategy of stacking multiple augmentations.
>
> > W2 & Q1: Placement of Baseline Comparisons
>
> I strongly suggest moving these results from the Appendix to the main text. For the paper to be convincing, it is crucial to prominently display empirical evidence where the proposed method outperforms strong baselines. Relegating these positive results to the Appendix diminishes their impact and makes the main text less persuasive.
>
> > Question 2: Data Augmentation (DA) vs. DML
>
> Please refer to my comments above under Weakness 1, where I discuss the relationship between DML and data augmentation in the context of the proposed experiment.
>
> > Minor Weaknesses: Structuring Assumption 1
>
> Regarding Assumption 1, my concern is primarily structural. Currently, the text reads more like a definition than a constraint. For mathematical precision, I recommend splitting it into two distinct parts:
> 1.	A Definition that formally describes "Approximate Translation-Invariance" (ATI) for fusion networks.
> 2.	An Assumption stating that the specific modality fusions employed in this work satisfy this ATI property.
>
> ---
>
> **General Recommendation on Revisions:** Finally, I would like to offer a piece of advice based on my experience as both a reviewer and an author. When reviewers request specific experiments or clarifications, it is because we identify them as critical for the completeness and persuasiveness of the work. Relegating these additions to the Appendix often diminishes their impact.
>
> I strongly suggest prioritizing the integration of reviewer-requested changes—not just mine, but those from all reviewers—directly into the main text whenever possible. Since the rebuttal revision allows for an extra page, I encourage you to utilize this space to move the critical experiments out of the appendices and into the body of the paper.

---

> ### Author Response · Authors · 2025-11-26
>
> We sincerely thank the reviewers for their constructive suggestions, which have significantly helped improve the quality and clarity of our manuscript. In response to the reviewers’ main concerns, we have revised the manuscript as follows:
>
> - **W1 - Clarification on DML and Variance:** We thank the reviewer for this helpful suggestion. We have revised **Section 3.3 (Lines 266–269)** to clarify that DML conceptually maximizes the variance of the kernel $\mathcal{K}$. We hope this revision adequately addresses the reviewer’s concern.
> - **W1, W2, Q1, Q2:**  We have moved key contents from Appendix C.2 and C.3 into the main paper on **Section 4.3 (Lines 338–370)**, presenting comparative results with data augmentation and flatness-aware approaches. Due to space limitations, we could not include all results in the main text; however, we report the main results demonstrating that **DML consistently improves performance**. We have also revised Appendix C.2 and Table results to highlight that incrementally adding DA techniques further improves performance, which partially supports our hypothesis.
> - **Minor Weakness:** We also appreciate the reviewers’ valuable suggestions regarding ATI. We have refined the **definition of the ATI property** and explicitly stated the **assumption that the given fusion network satisfies ATI** in **Section 3.1 (Lines 146–153).**
>
> &nbsp;
>
> We sincerely appreciate the reviewer’s suggestion and general recommendation. For major revisions, we will prioritize updating the main paper rather than the appendices. We again appreciate the reviewer's valuable feedback and insightful comments.

---

> ### Comment · Reviewer_LpsV · 2025-11-28
>
> I acknowledge the improvements made to the manuscript and am raising my score to Weak Accept.
>
> **Note on Assumption 1:** The current text lists specific fusion methods (additive, concatenation, MLPs, attention). To maintain rigor, you should only list these examples if you can demonstrate that they represent Approximate Translation-Invariant (ATI) functions. Otherwise, please remove the specific examples to ensure the assumption remains mathematically sound.
>
> **Note on Score Update:** I intend to raise my score to Weak Accept as stated above. However, OpenReview is currently restricting edits to the score field, despite the modification deadline being December 2nd. It appears this is a known system-wide issue affecting many reviewers. I will monitor the platform and formally update the score as soon as the technical restriction is lifted.

---

> > ### Author Response · Authors · 2025-11-28
> >
> > **Dear Reviewer LpsV**,
> >
> > Thank you very much for your kind suggestions and constructive discussion during the rebuttal. We sincerely appreciate the time and effort you devoted to reviewing our work, which has significantly improved our paper.
> > Moreover, according to the suggestions, we will revise the manuscript, particularly regarding **Note on Assumption 1**.
> >
> > Lastly, we are grateful for your decision to raise the score.
> >
> > &nbsp;
> >
> > Best regards,
> >
> > The Authors of Submission 12001

---

### Official Review · Reviewer_9cP9 · 2025-11-01

**Soundness:** 2
**Presentation:** 2
**Contribution:** 2
**Rating:** 4
**Confidence:** 3

**Summary:**

This paper presents a novel theoretical perspective explaining why multimodal learning (MML) typically outperforms unimodal learning. The authors' core argument is that multimodal learning introduces a **“Convolutional Smoothing”** effect, leading to a flatter loss landscape, which correlates with improved generalization and robustness.

Building on this theory, the paper further proposes a training strategy called **“Distributional Multimodal Learning” (DML)**. This approach avoids fixed modality pairings during training, instead performing random pairings within the same hypothesis space.

Experimental results across multiple datasets (e.g., Kinetics-Sounds, AVMNIST) demonstrate that multimodal learning (MML) indeed yields a flatter loss landscape than unimodal learning, while DML achieves an even flatter landscape than MML, delivering higher performance and robustness.

**Strengths:**

1. Establishing a link between MML and loss landscape smoothness, and attempting to provide a mathematical explanation through **“convolutional smoothing”** (Theorem 1 and Theorem 2) offers a novel perspective.

2. The paper provides substantial experimental evidence (performance, robustness, loss visualization, Hessian eigenvalues) supporting its core claim: MML → smoother landscape → better performance.

3. DML is a simple, intuitive, and effective strategy that consistently outperforms standard SML (fixed pairing) across all experiments.

**Weaknesses:**

1. The paper's theoretical foundation rests on overly strong assumptions. Its theory is based on the **Assumption 1 (Approximate Translation-Invariance of Fusion Networks)**. In Appendix C.2, the authors themselves acknowledge that this assumption is only a first-order Taylor approximation for non-linear MLPs and may not hold for attention mechanisms. This could render the theoretical framework fundamentally inapplicable to current state-of-the-art fusion models based on Transformers and Cross-Attention.


2. Suspected paste error on page 7, lines 351-352: “...we additionally report the ratio between **thewe also report the ratio between the** largest a”

3. Bold formatting in Table 2 is confusing: some rows have both Modality i and Modality j with bolded values (**CREMA-D**), while others do not. Notably, for **UPMC-Food101**, the value for **λ∗ under Modality j** is 1.4935, which is the best in its column but is not bolded.

**Questions:**

1. The proposed DML (Random Pairing) is analogous to a data augmentation technique. The improved results from DML may stem from the data augmentation effect of random pairing, which significantly increases the number of training pairs, rather than the theoretical mechanism of distributional alignment.

2. **Assumption 2** requires xi and xj to be conditionally independent given y. Under this independence, SML and DML should sample from the same distribution. SML samples from the true joint conditional probability distribution p(x_i,x_j|y), while DML independently samples from the marginal conditional probability distributions p(x_i|y) and p(x_j|y). If **Assumption 2** holds, then p(x_i|y)p(x_j|y)=p(x_i,x_j|y), meaning the sampling distributions are identical. If SML and DML share the same distribution, their results should be similar. However, experiments show DML significantly outperforms SML. This suggests the conditional independence assumption may not hold in practice, which could challenge the theoretical framework.

---

> ### Author Response · Authors · 2025-11-20
> **Response to Reviewer 9cP9 (Part 1/3)**
>
> ### **Dear Reviewer 9cP9**,
>
> We thank the reviewer for their insightful comment, which highlights a key ambiguity in our current theoretical discussion. During the rebuttal period, we respectfully believe that several core concerns can be addressed as outlined below:
>
> &nbsp;
>
> >### **Weakness 1. Justification and Validity of Assumption 1 (ATI)**
>
> The reviewer correctly identifies that our current Appendix C.2 analyzes the linear approximation primarily from a **finite-dimensional perspective**. We concede that this view alone, as currently written, may appear insufficient to fully justify the assumption for strongly non-linear mechanisms like cross-attention.
>
> Yet, the theoretical foundation can be naturally extended by considering the **infinite-width limit of modern over-parameterized architectures [1,2,3]**, a standard paradigm for analyzing contemporary deep networks. In this "lazy training" (NTK) regime, it is a well-established theoretical result that even complex architectures like **Transformers including attention mechanism can be linearly approximated** (i.e., they are dominated by their first-order Taylor expansion). Prior work has shown this explicitly for attention layers [1, 2], and our current finite-dimensional analysis is consistent with this broader framework.
>
> **Testing of Assumption 1 for cross-attention fusion networks**
>
> | Feature Norm (Scale) | Invariance Error | Error Percentage |
> | --- | --- | --- |
> | 0.4267 | 0.0051 | 1.1952 % |
> | | |
>
> Thus, Assumption 1 is still a valid approximation backed by this extensible theoretical framework. To further validate this theoretical argument, we have added new experiments using a **cross-attention fusion network.** As shown in the upper table, it also shows a minimal **approximation error**, empirically confirming our assumption's validity.
>
> We still acknowledge that our submission did not make this theoretical connection sufficiently clear, where we only viewed in finite dimension space. **For clarity, we moved Appendix C.2 to Appendix D.1, as discussion part, to explicitly include this crucial discussion on the infinite-dimensional justification and added Cross-Attention part**, clarifying how our assumption 1 can be extended to complex networks.
>
> **[1]** Yang, G. "Tensor Programs II: Neural Tangent Kernel for Any Architecture." ICLR 2021.\
> **[2]** Hron, J., et al. "Infinite Attention: NNGP and NTK for Deep Attention Networks." ICML 2020.\
> **[3]** Jacot, A., et al. "Neural Tangent Kernel." NeurIPS 2018.
>
> &nbsp;
>
> >### **Weakness 2 & 3 ⇒ Minor Corrections**
>
> We thank the reviewer for pointing out these mistakes. We corrected the paste error and fixed the inconsistent bold formatting in Table 2 (currently Table 4) in the revised manuscript.

---

> ### Author Response · Authors · 2025-11-20
> **Response to Reviewer 9cP9 (Part 2/3)**
>
> >### **Question 1. DML and Data Augmentations**
>
> We sincerely thank the reviewer for their insightful comment regarding the relationship between our proposed DML and conventional Data Augmentation (DA). The reviewer rightly points out that DML increases the number of training pairs, an effect superficially similar to DA. Yet, we would like to respectfully offer two perspectives to clarify the distinction and novelty of our approach: the conventional concept of DA and its practical effects.
>
> First, we would like to emphasize the difference between DML and traditional DA. Conventional DA operates on the input space, applying semantics-preserving perturbations (e.g., cropping, rotation, jitter for visual data), where its goal is to expand the distribution within the original semantic boundaries. In contrast, our DML (Random Pairing) operates on the hypothesis space (i.e., label space). DML generates pairs by matching instances that share the same hypothesis but originate from different data points.
>
> For example, consider two pairs: `[Poodle_image, Poodle_audio]` and `[Bichon_image, Bichon_audio]`, **both labeled as "dog."** Traditional DA can only perturb these two pairs individually and cannot generate a pair like `[Poodle_image, Bichon_audio]`. DML, however, can create such a novel pair by leveraging shared labels. This inter-modality stochasticity encourages the model to learn more diverse semantics. Moreover, DML is complementary to conventional DA. One could first generate `[Poodle_image, Bichon_audio]` via DML, and then apply standard DA (e.g., cropping, jittering) to this new pair. This demonstrates that DML introduces a mechanism fundamentally distinct from, yet compatible with, traditional DA.
>
> We provide additional experiments that apply DA on the image modality. As a result, simply applying data augmentation shows a slight improvement on both SML & DML. This shows that how DA is complementary with DML.
>
> | Data Augmentation | KS | AVMNIST | CREMA-D | Food101 |
> | --- | --- | --- | --- | --- |
> | SML | 62.95 | 70.35 | 59.62 | 91.48 |
> | + Color Jitter | 63.87 | 70.39 | 59.66 | 91.95 |
> | + Cropping | 63.19 | 70.59 | 59.1 | 92.13 |
> | + Horizontal Flipping | 64.14 | 70.87 | 59.35 | 91.99 |
> | DML | 65.89 | 71.69 | 59.77 | 92.84 |
> | + Color Jitter | 66.49 | 71.22 | 59.94 | 93.57 |
> | + Cropping | 65.77 | 71.8 | 60.57 | 95.21 |
> | + Horizontal Flipping | 66.85 | 72.07 | 60.51 | 94.29 |
>
> Second, even if one interprets DML broadly as a form of DA, this perspective still supports our core novelty, the smoothing perspective. Recent work [1] shows that DA is not merely a method to increase data quantity but also induces a smoothing effect. Specifically, according to recent work, DA can be interpreted as introducing perturbations in the parameter space, guiding the model toward flatter minima and tighter generalization bounds. (DA → perturb parameter space → flatter minima / tighter generalization bound)
>
> Viewed through this lens, DML effectively smooths the loss landscape in a manner analogous to DA. This theoretical interpretation reinforces our claim that DML promotes more robust and generalizable solutions, highlighting its novelty from both a mechanism and theoretical perspective.
>
> Therefore, while DML shares the high-level goal of improving generalization with DA, its hypothesis-space pairing mechanism across modalities is fundamentally distinct and complementary to the conventional definition of DA. Moreover, framing DML amplifies the smoothing effect, aligns with modern theoretical insights on generalization, and further keeps our novelty.
>
> We thank the reviewer again for this valuable feedback and will revise the manuscript to clarify this distinction. **We revised our manuscript on Appendix C.2.**
>
> [1] Yoo, Weebum, and Sung Whan Yoon. "A Flat Minima Perspective on Understanding Augmentations and Model Robustness." Preprint (2025)
>
> &nbsp;
>
> ----
>
> ### **Revision of Data Augmenation (DA)**
>
> We have revised the DA section in the manuscript by including additional experimental results on incrementally adding DA techniques to both SML and DML. Specifically, we added **Section 4.3** in the main manuscript and updated **Appendix C.2** to reflect experiments with additional DA techniques. These results support our hypothesis that increasing data diversity, via DML or DA, leads to improved performance and further validates the effectiveness of our main approach.
>
> **Again, we thank the reviewer for the constructive feedbacks.**

---

> ### Author Response · Authors · 2025-11-20
> **Response to Reviewer 9cP9 (Part 3/3)**
>
> >### **Question 2. Discussion on Assumption 2**
>
> The reviewer’s argument—that Assumption 2 would make SML and DML effectively identical—implicitly assumes that the SML (Fixed Pair) dataset is a representative sampler of the true joint conditional distribution $p(x_i, x_j \mid y)$. We respectfully clarify that this assumption does not hold, and this distinction is central to the observed performance difference.
>
> The key distinction is that the SML sampling space constitutes a strict subset of the DML sampling space. As the reviewer notes, while DML samples from the full conditional product of marginals $p(x_i \mid y) p(x_j \mid y)$ covering **all valid cross-modal pairings** consistent with label hypothesis $y$, SML (Fixed Pair) samples only from the original paired observations in the dataset, excluding all cross-pairings between modalities that do not co-occur in the raw data, even though such combinations are valid members of $p(x_i \mid  y)p(x_j \mid  y)$ under Assumption 2.
>
> Consequently, both methods sample from the same underlying distribution $p(x_i \mid  y)p(x_j \mid  y)$ under Assumption 2. However, SML's sampling procedure is restricted to a narrow subset of this space (only co-occurring pairs), while DML explores it more broadly (all valid combinations). This difference in sampling coverage—not a difference in the target distribution—explains the performance gap.
>
> This distinction is fully consistent with our theoretical analysis (Theorem 2, Remarks 1 & 2). SML’s limited sampling induces a kernel with minimal variance (e.g., expectation of dirac delta function), yielding a weak smoothing effect. In contrast, DML’s higher-variance sampling activates a much stronger smoothing effect, enabling better generalization and reducing reliance on spurious correlations.
>
> Therefore, DML does not violate Assumption 2; instead, it leverages the full conditional product space, whereas SML samples only a small, low-variance subset. This intrinsic difference in sampling distributions fully explains the observed performance gap.
>
> Again, we sincerely thank the reviewer for raising this important point. We have revised our manuscript and included a detailed discussion in Appendix D.2.
>
> &nbsp;
>
> ==========================================================
>
>
> **We are again grateful for the reviewer’s insightful feedback. We hope that our responses have clarified the key points and addressed the concerns raised.**

---

### Author Response · Authors · 2025-11-20
**Overview of Manuscript Revisions**

(Revised)

Dear Reviewers,

We truly appreciate your constructive and insightful feedback. We have carefully revised the original paper based on your thoughtful comments. All revised text in the manuscript has been marked in $\textcolor{teal}{\textbf{teal}}$ for the clarity:

### **For Reviewer 9cP9**

- We have updated $\textcolor{teal}{\textbf{Table 13}}$ to justify Assumption 1 and added a section on Cross-Attention to demonstrate linearity in complex networks. This content has been relocated from Appendix C.2 to $\textcolor{teal}{\textbf{Appendix D.1}}$. (**Weakness 1**)
- We have corrected typos in $\textcolor{teal}{\textbf{lines 351-352}}$ and $\textcolor{teal}{\textbf{Table 2}}$. (**Weaknesses 2, 3**)
- We have added and again revised a section on Data Augmentation (DA) in $\textcolor{teal}{\textbf{Appendix C.2}}$. (**Question 1**)
- We have added $\textcolor{teal}{\textbf{Appendix D.2}}$ to include a discussion regarding Assumption 2. (**Question 2**)
- We have added $\textcolor{teal}{\textbf{Section 4.3}}$  to present comparisons between DML and Data Augmentation as well as flatness-aware approaches. (**Question 1**)

### **For Reviewer LpsV**

- We have addressed **Weakness 1: Weak connection between theory and method** in an *Official Comment*.
- We have revised the main manuscript in $\textcolor{teal}{\textbf{Lines 266–269}}$ to clarify and support the superiority of DML.
- We have added $\textcolor{teal}{\textbf{Section 4.3}}$ to present comparisons between DML and Data Augmentation as well as flatness-aware approaches. (**Weakness 2, Question 1 & Question 2**)
- We have added and again revised $\textcolor{teal}{\textbf{Appendix C.3}}$ to include additional comparison between flatness-aware approaches and DML. (**Weakness 2 & Question 1**)
- We have added and again revised $\textcolor{teal}{\textbf{Appendix C.2}}$ for additional discussion and experiments on Data Augmentation in  (**Question 2**)
- We revised Assumption 1 by separating it into a formal definition of ATI and a new assumption that the fusion network satisfies the ATI property on $\textcolor{teal}{\textbf{Lines 146–154}}$ . (**Minor Weakness**)

### **For Reviewer EYm2**

- We have provided an *Official Comment* regarding the concern on **Weakness 3: Intra-class random perturbation**.
- We have added $\textcolor{teal}{\textbf{Appendix D.2}}$ for a detailed discussion on the validity of Assumption 2. (**Weakness 1**)
- We also updated $\textcolor{teal}{\textbf{Appendix D.1}}$ (previously Appendix C.2) to provide justification for Assumption 1. (**Weakness 2**)
- We have added $\textcolor{teal}{\textbf{Appendix C.4}}$ to present the performance results on datasets with three modalities. (**Weakness 4**)

### **For Reviewer Hoxa**

- We have addressed the primary concerns regarding **Weakness 1 and Weakness 2** in an *Official Comment*, referencing relevant sections of our manuscript.
- We have included a detailed analysis and justification for Assumption 2 in $\textcolor{teal}{\textbf{Appendix D.2}}$. (**Question 1**)


&nbsp;

Again, we sincerely thank all reviewers for the constructive discussions and insightful comments. We truly appreciate your efforts, and we believe these feedbacks and discussions have significantly strengthened this work.

---

### Author Response · Authors · 2025-12-02
**Summary of Rebuttal Responses and Final Comments**

Dear PC, SAC, AC, and Reviewers

We sincerely thank the PC, SAC, AC, and all reviewers for their effort, and dedication in reviewing our manuscript.Due to uncertainties from the information leakage issue, we sincerely thank the reviewers for their insightful comments, and both the previous and new ACs for ensuring a smooth continuation of the review process.

Below, we summarize the **current status and our responses to the major concerns**:

---

**Current Status:**

- **Reviewer 9cP9 (Initial Rating 4)**: No additional comments were received after the rebuttal.
- **Reviewer LpsV (Initial Rating 4 → 6)**: A discussion took place during the rebuttal period, after which the *reviewer indicated an intention to raise the score*.
- **Reviewer EYm2 (Initial Rating 4):** The reviewer raised additional concerns regarding Assumption 2. *We then provided clear and detailed clarification in our response*.
- **Reviewer Hoxa (Initial Rating 8):** No additional comments were received after the rebuttal.

---

Most concerns raised by the reviewers have been addressed, particularly through detailed discussions with **Reviewer LpsV**. The **main concerns** raised by **Reviewer LpsV** were as follows:

- **Concern 1:** The connection between our theoretical framework and DML’s practical effect on smoothness, which is closely related to the justification of Assumption 2 (also from **Reviewers 9cP9, EYm2, and Hoxa**).
- **Concern 2:** Comparisons with Data Augmentation (DA) and beyond (also raised by **Reviewers 9cP9 and LpsV**), and flatness methods.

We then clairfy this as follows:

- **Response 1 (DML vs. SML):**

    We clarified that ***SML exhibits low variance in the kernel $\mathcal{K}$***, behaving similarly to a Dirac delta distribution, whereas ***DML explicitly maximizes kernel variance*** (i.e., increases entropy), resulting in a ***smoother loss landscape as formalized in Theorem 2 and Remark 2***. We further emphasized that this formulation ***directly leverages Assumption 2 by broadening distributional coverage*** via sharing hypothesis space $y$, whereas SML typically restricts exploration to a limited, low-variance subset of the distribution. These clarifications have been incorporated into **Section 3.3 (Lines 266–269)**.

- **Response 2 (Comparison with DA and Flatness-Aware Methods):**

    We clarify that ***DA operates in the input space***, applying semantics-preserving perturbations while ***DML operates in the hypothesis space***, matching samples with the ***same hypothesis space $y$*** across different instances, and ***leads to complementary effect***. We then show that applying DA on DML yields additional performance gains on **Section 4.3 and Appendix C.2.** In last, we also have provided results that DML outperform against representative flatness methods, including LS, SWA, and SAM, all applied to SML across our benchmarks.

---

In addition to the points discussed above, **additional major concerns** from reviewers have been made as below:

- **Validity and Justification of Assumptions 1 - Approximated Translation Invariance (ATI) (Reviewer 9cP9, EYm2)**:
    - **Theoretical Justification:** We provided theoretical justification that common fusion networks satisfy ATI (**Appendix D.1**).
    - **Empirical Validation (Minimal Linearity Gap):** We also reported empirical results on **Appendix D.1** showing a negligible linearity gap, supporting the approximation.

---

Moreover, we clarified **individual concerns** as follows:

- **Reviewer 9cP9 (Initial Rating 4)**
    - **Minor Concerns:** Revised our manuscript typos and bold formatting in Table 2 (currently Table 4)
- **Reviewer LpsV (Initial Rating 4 → 6)**
    - **Minor Weakness:** Reformulated the original Assumption 1 into two parts—Definition 1 (Approximated Translation Invariance) and Assumption 1 (Fusion Networks Satisfy ATI)—to improve clarity and formal distinction.
- **Reviewer EYm2 (Initial Rating 4)**
    - **Intra-class random perturbation (W.3):** Clarifying that DML does not arbitrarily match unimodal data; ***it operates on shared hypotheses and improves performance by expanding the effective hypothesis space***, not by generating new data.
    - **Extra modalities (W.4):** Provide additional 3-modality experiments in **Appendix C.4**, demonstrating that DML consistently improves both accuracy and flatness metrics.
- **Reviewer Hoxa (Initial Rating 8)**
    - **Architectures and Scalability Issue (W.1, W.2):** Clarifying that **Appendix C.1** already provides validation using CLIP-based architectures, confirming that the method remains effective at scale.

---

Lastly, ***we summarized the manuscript revision on below official comment***. Once again, we would like to express our sincere gratitude to the PC, SAC, ACs, and all reviewers for their insightful feedback and continuous support. We truly appreciate the opportunity to strengthen our work through this process.

Sincerely,

The Authors

---

### Meta-Review · Area_Chair_pZ67 · 2026-01-04

**Summary:**

This paper explains the advantage of multimodal learning from the perspective of loss landscapes. It proposes convolutional smoothing and shows that stochastic modality pairing yields flatter loss landscapes and improved generalisation. The main concerns include the unrealistic assumptions, unclear difference between DML and data augmentation (e.g., CutMix) and lack of empirical studies on large-scale datasets and models. After reading the paper and the review, I think the concerns on theoretically assumptions and novelty are still valid. Specifically, the rebuttal acknowledges that the paper only considers the lazy regime without considering the rich regime. The difference between DML and CutMix is not explained well. In summary, I would recommend rejection for the submission.

**Reviewer Concerns:**

The following concerns are addressed.

1.	There are writing errors (9cP9)
2.	The connection between theory and method is weak (LpsV)
3.	Lack of comparison with baselines (LpsV)
4.	Performance gain of DML is counterintuitive (EYm2)
5.	It is unclear how to extend DML to three or more modalities (EYm2)
6.	More and larger-scale experiments are needed to validate the robustness (Hoxa)

The following concerns are outstanding.

1.	Assumptions are not realistic (9cP9, EYm2, Hoxa)
2.	Connection to data augmentation is not clear (LpsV, 9cP9)

**Reviewer Scores:**

Reviewer 9cP9 will keep 4.

Reviewer LpsV has agreed to increase to 6.

Reviewer EYm2 will keep 4.

Reviewer Hoxa will keep 8.

---

### Decision · Program_Chairs · 2026-01-26

Reject